Ecological and Evolutionary Science

# Streamlined and Abundant Bacterioplankton Thrive in Functional Cohorts

Rhiannon Mondav,[a] Stefan Bertilsson,[a,b,c] Moritz Buck,[a,c,d] Silke Langenheder,[a] Eva S. Lindström,[a] Sarahi L. Garcia[a,e]

[a]Department of Ecology and Genetics, Limnology, Uppsala University, Uppsala, Sweden
[b]Science for Life Laboratory, Uppsala University, Uppsala, Sweden
[c]Swedish University of Agricultural Sciences, Uppsala, Sweden
[d]National Bioinformatics Infrastructure Sweden, Uppsala, Sweden
[e]Department of Ecology, Environment, and Plant Sciences, Science for Life Laboratory, Stockholm University, Stockholm, Sweden

**ABSTRACT** While fastidious microbes can be abundant and ubiquitous in their natural communities, many fail to grow axenically in laboratories due to auxotrophies or other dependencies. To overcome auxotrophies, these microbes rely on their surrounding cohort. A cohort may consist of kin (ecotypes) or more distantly related organisms (community) with the cooperation being reciprocal or nonreciprocal and expensive (Black Queen hypothesis) or costless (by-product). These metabolic partnerships (whether at single species population or community level) enable dominance by and coexistence of these lineages in nature. Here we examine the relevance of these cooperation models to explain the abundance and ubiquity of the dominant fastidious bacterioplankton of a dimictic mesotrophic freshwater lake. Using both culture-dependent (dilution mixed cultures) and culture-independent (small subunit [SSU] rRNA gene time series and environmental metagenomics) methods, we independently identified the primary cohorts of actinobacterial genera "*Candidatus* Planktophila" (acI-A) and "*Candidatus* Nanopelagicus" (acI-B) and the proteobacterial genus "*Candidatus* Fonsibacter" (LD12). While "*Ca.* Planktophila" and "*Ca.* Fonsibacter" had no correlation in their natural habitat, they have the potential to be complementary in laboratory settings. We also investigated the bifunctional catalase-peroxidase enzyme KatG (a common good which "*Ca.* Planktophila" is dependent upon) and its most likely providers in the lake. Further, we found that while ecotype and community cooperation combined may explain "*Ca.* Planktophila" population abundance, the success of "*Ca.* Nanopelagicus" and "*Ca.* Fonsibacter" is better explained as a community by-product. Ecotype differentiation of "*Ca.* Fonsibacter" as a means of escaping predation was supported but not for overcoming auxotrophies.

**IMPORTANCE** This study examines evolutionary and ecological relationships of three of the most ubiquitous and abundant freshwater bacterial genera: "*Ca.* Planktophila" (acI-A), "*Ca.* Nanopelagicus" (acI-B), and "*Ca.* Fonsibacter" (LD12). Due to high abundance, these genera might have a significant influence on nutrient cycling in freshwaters worldwide, and this study adds a layer of understanding to how seemingly competing clades of bacteria can coexist by having different cooperation strategies. Our synthesis ties together network and ecological theory with empirical evidence and lays out a framework for how the functioning of populations within complex microbial communities can be studied.

**KEYWORDS** *Actinobacteria*, alphaproteobacteria, aquatic, bacterioplankton, common goods, ecology, evolution, metagenomics, microbial communities, networks

Address correspondence to Rhiannon Mondav, rhiannon.mondav@gmail.com.

Defining and dividing a niche by auxotrophies and predator evasion

The stable coexistence of species is described by a continuum of interspecific symbiotic interactions from parasitism through neutralism to mutualism (1). The extended coexistence of microbial strains is described by intraspecific relationship models limited to niche theory (ecotype differentiation) (2) and predator control of dominant strains (3). Combining these ecological theories of coexistence with phylogenetic information can produce a simplified model with a new holistic perspective for understanding aquatic microbial communities. In aquatic microbial communities, especially for non-particle-associated bacterioplankton, physical structures and proximity as seen in soil, host-associated microbiome, and biofilm communities (4) are absent. As cooperation models have been developed from the study of such tightly interwoven assemblages, their models may not be the most suitable. Also absent is the habitat isolation which limits dispersal as seen in, e.g., soil ecosystems. In aquatic systems, interactions between nutrients and biota can occur from micro- up to macroscale distances involving stochastic effectors from Brownian motion up to whole-system circulation currents. Direct but distant interactions are therefore difficult to identify based purely on proximity as common goods and toxins in the environment can come from multiple sources and be transported long distances. Data from long-term monitoring of aquatic environments can be used to test the validity of ecological models.

Lake Erken in Sweden has been extensively studied for several decades, and ample background information on nutrient cycling, planktonic communities, and biogeochemical processes combined with well-established infrastructure makes this ecological observatory an excellent choice for the study of competing microbial assemblage processes (5–7). Several studies have investigated the microbial component of this lake ecosystem (8–13), revealing that the most abundant microbes in Lake Erken belonged to the LD12 (now described as the candidate genus "*Candidatus* Fonsibacter" [14]) and acl (ACK-M1, hgcl, but now described as the candidate order *Nanopelagicales*) clades (11, 12, 15–20). "*Ca.* Fonsibacter" is a nonmarine genus within the ubiquitous and abundant *Pelagibacteraceae* family which includes the marine *Pelagibacter* genus (SAR11). *Nanopelagicales* is a freshwater order with two described genera, "*Candidatus* Planktophila" and "*Candidatus* Nanopelagicus." Both "*Ca.* Fonsibacter" and the *Nanopelagicales* have small cell size, reduced genomes (1.16 to 1.48 Mbp), multiple auxotrophies, a requirement for reduced sulfur compounds, and rhodopsins and have been recalcitrant to maintenance in axenic laboratory culture (14, 21–23). "*Ca.* Fonsibacter" and "*Ca.* Nanopelagicus" genera both have additional amino acid auxotrophies (23, 24). Recently, a single strain of "*Ca.* Fonsibacter" and two strains of the *Nanopelagicales* genus "*Ca.* Planktophila" (acl-A and acl-A4) were successfully maintained in axenic cultures by customized liquid media in the case of "*Ca.* Fonsibacter" (14) and the addition of an active enzyme to filtered lake water media for "*Ca.* Planktophila" (25). While both genera have been described and cultured, neither have been deposited in culture collections and so retain *Candidatus* status. The difference in cultivation methods point toward a difference in lifestyles which would also be seen in the evolution and ecology of these two abundant and ubiquitous freshwater bacterioplankton clades.

The ubiquity of the small-genome bacterial lineages, such as *Nanopelagicales* and "*Ca.* Fonsibacter" in freshwaters, is thought to be a consequence of their streamlined genomes and small cell size (26). On the one hand, small cell size increases the surface area-to-volume ratio, reducing diffusion bottlenecks, thereby decreasing foraging time in these nonmotile organisms (4). On the other hand, reduced genomes have lost extraneous metabolic functions while having very low gene redundancy, reduced regulatory components, and high coding density (27). This combined streamlining and shrinkage reduce the energetic and material cost of genome, proteome, and cellular maintenance. Because their small genomes encode fewer metabolic functions than average, it has been proposed that they require a large effective population (large $N_e$) within which ecotypes diverge to provide the genomic flexibility usually seen in individual cells of bacteria with larger genomes. A large $N_e$ is also required for purifying selection to be efficient at reducing and maintaining small genome size. Such streamlined microbes were first noted in low-nutrient aquatic environments and streamlining

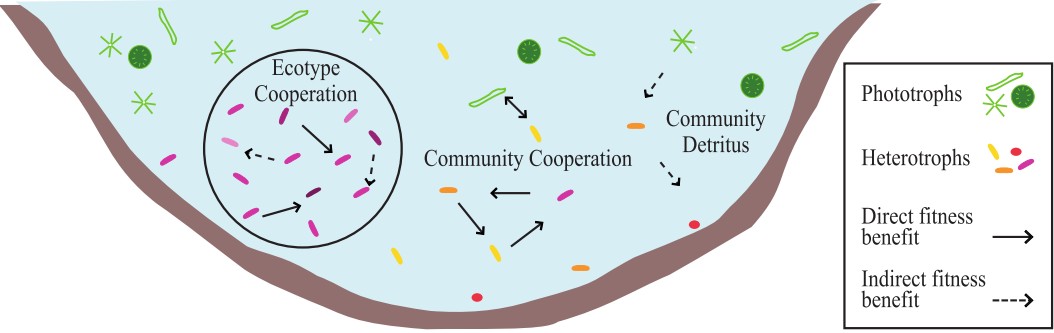

**FIG 1** Cooperation models shown in a lake setting. The ecotype cooperation model describes anabolic variability within species as the main contributor to coexistence while allowing for recombination and swapping of metabolic modules across strains. The community cooperation model also encompasses anabolic complementarity but between different species and includes common goods ranging from expensive to costless. The community detrital model places complementarity both between and within species but describes catabolic variation.

has since then been suggested as being of selective advantage under reduced nitrogen availability (28). However, *Nanopelagicales* and "*Ca*. Fonsibacter" clades are dominant and successful in nonmarine environments such as Lake Erken, which is currently mesotrophic heading toward eutrophication. Further, as Lake Erken is ice and snow covered most winters (temperature and light limiting) and stratified in summer (thermo- and oxyclines), it is not a stable environment where organisms with reduced regulatory capacity are expected to thrive. It is unlikely that dominance in such productive lakes is made possible purely by streamlining, as the subsequent loss of genomic flexibility required to respond to a dynamic and variable habitat such as Lake Erken would counteract the benefits. Furthermore, large populations will attract predators, though both clades may escape some grazing pressure due to their small size (29, 30). "*Ca*. Fonsibacter" may additionally have "slippery" membrane properties similar to its sister marine lineage SAR11 (31) to further evade grazing. Additionally, all three genera encode cell membrane modification genes in hypervariable regions predicted to assist escape from viral predation (23, 32). A large $N_e$ where the divergence is focused in immunogenic or predator escape mechanisms (defense specialist) will help a species survive population sweeps caused by predation, but it does not explain the intra- or interspecific diversity or the apparent lack of competition between these three genera.

We combined the continuum of symbiotic interactions with phylogenetic information to produce a simplified model with a new holistic perspective for understanding ecological strategies in aquatic microbial communities (Fig. 1). The ecotype cooperation model describes how anabolic variability within a species population, i.e., strain level variation, enables all the strains working together to produce nutrients required for cellular function and replication. It also allows for recombination and swapping of metabolic modules across strains. The community cooperation model also has a basis in anabolic complementarity but places the metabolic variation within a symbiotic group of different species whereby cooperators provide expensive (Black Queen hypothesis) or costless common goods (33, 34). The community detrital model is similar to the community cooperation model but focuses on uptake and catabolism of nutrients released by cellular death mediated by toxicity, starvation, or predation. Aquatic habitats in particular, due to transport and diffusion, are understood to have a unique niche for detritivores that may select for streamlined microbes (35).

Here we examine an 8-year 16S rRNA amplicon time series along with 26 dilution mixed cultures and eight lake metagenomes from the same lake (Fig. 2) to identify putative supporting microbes (cohort) and assess whether that support is best described by ecotype, community, and/or detrital models. We began with 16S rRNA operational taxonomic unit (OTU) network modeling (to define a candidate cohort that might provide for dependencies of the three target genera), then examined the phylogenetic relatedness of each cohort (to determine whether there was a signal for

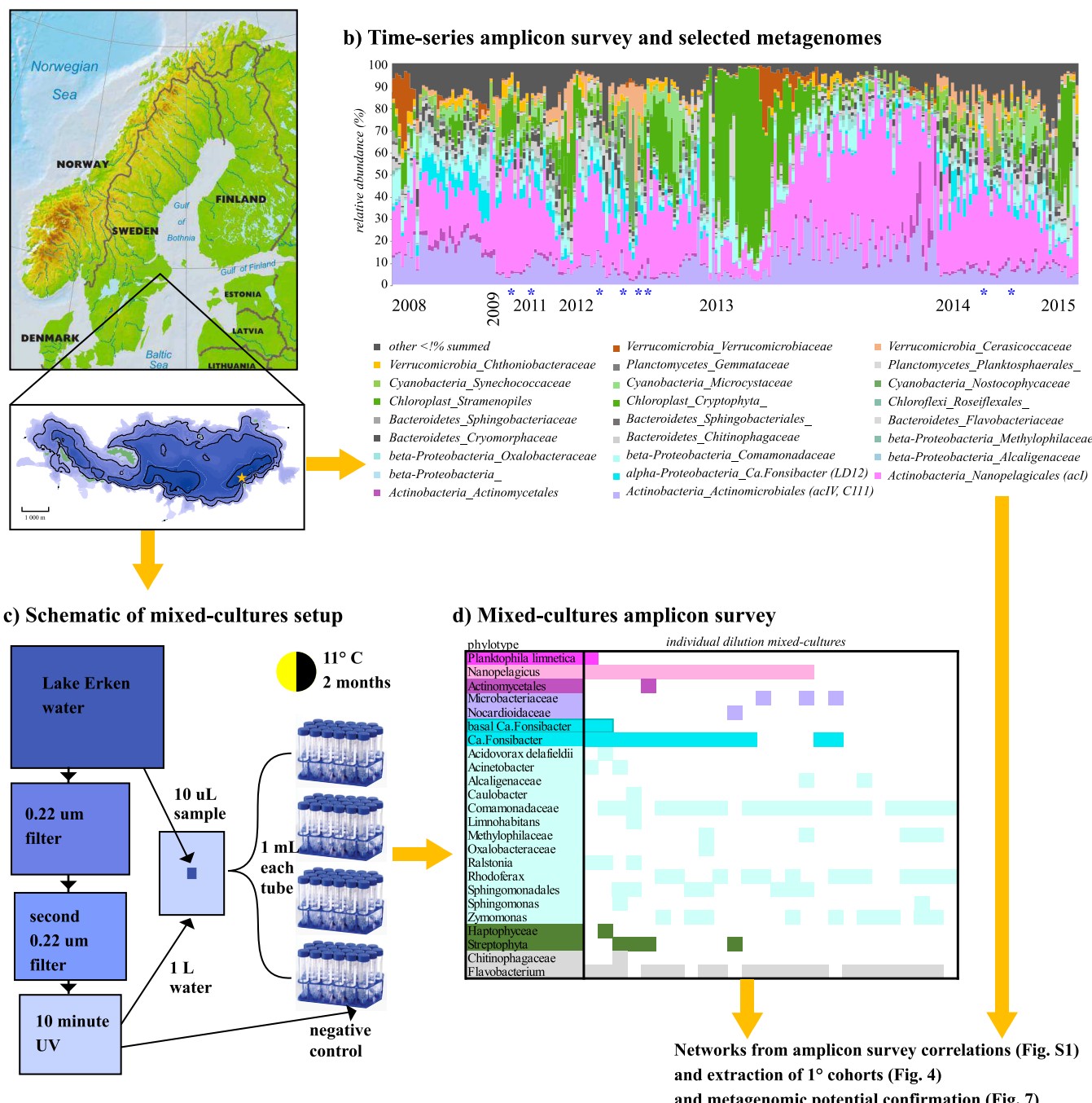

**FIG 2** (a) Location of Lake Erken and the sampling site within the lake. (b) Relative abundance of families from community time-series amplicon survey with metagenome-sequenced samples marked by blue asterisks. (c) Schematic for dilution mixed-cultures. Mixed-culture samples were taken in March 2016, the first spring thaw after the amplicon data set was sequenced. (d) Presence/absence of families detected in mixed-cultures via amplicon survey.

ecotype or community), and reviewed the metabolic diversity of the target group (to establish its dependencies and diversity) through publications and genome analysis. Finally, we investigated potential providers of common goods, including the catalase-peroxidase (KatG) (a known dependency of some *Nanopelagicales* [25]), B-group vitamins, and cyanophycin production through database searches and within the metagenomes. Our proposed framework focuses more on neutral to positive interactions, as the laboratory experimental design does not permit detection of negative interactions even though the time series exposed potential negative interactions.

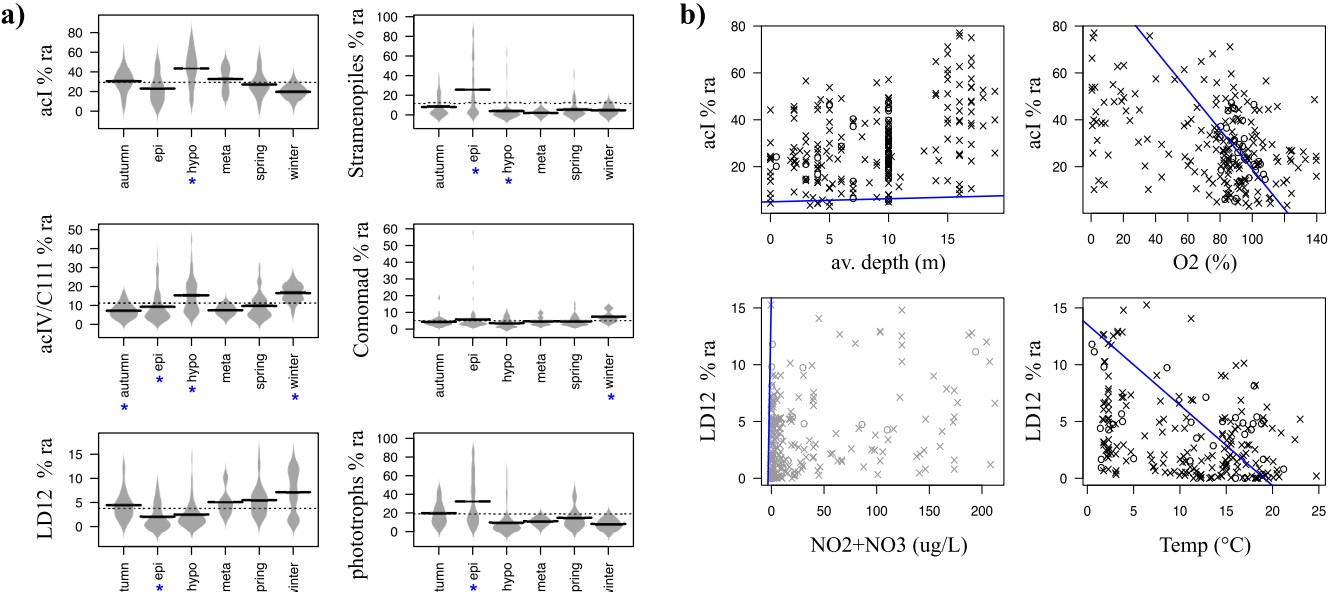

**FIG 3** (a) Associations between lake cycle/season and dominant taxa in Lake Erken. The *x* axes show three seasons (spring, autumn, and winter) and three lake layers of summer stratification (epi-, hypo-, and metalimnion). The *y* axes show the relative abundance (ra) of the six most dominant taxa (acl, *Stramenopiles*, acIV, *Comamonadaceae*, LD12, and grouped phototrophs). Seasons/layers where ra was significantly different (KW *P* < 0.001, KWmc *P* < 0.001) to at least two others are denoted by blue asterisks. (b) Correlations between taxon relative abundance (percent ra) and environment parameters, only showing those with linear (Pearsons) correlation (*r* > |0.3|, *P* < 0.001). Data points with "×" are 2011 to 2015 MiSeq data, while circles show 454 data from 2008. Linear correlations are shown by blue lines.

## RESULTS

**Time series taxon abundance was poorly correlated with environmental parameters.** A general overview of abundance and prevalence of taxa can provide insights into microbial ecosystems. The most abundant clades detected in the small subunit (SSU) rRNA gene time series were the actinobacterial order *Nanopelagicales* at 29% average relative abundance (av.ra), the eukaryote phytoplankton genus *Stramenopiles* 12% av.ra, the actinobacterial clade "acIV" (C111) 11% av.ra, the betaproteobacterial family *Comamonadaceae* 5% av.ra, the alphaproteobacterial candidate genus "*Ca.* Fonsibacter" 4% av.ra, and the verrucomicrobial family *Cerasicoccaceae* 3% av.ra (Fig. 2b).

Correlations between measured environmental parameters and taxon abundance can provide evidence of linkages between species dominance and their physical environment. There were few significant differences in the relative abundance of dominant clades across lake cycle/season (Kruskal-Wallis [KW] with *post hoc* testing [Kruskal-Wallis multiple comparison {KWmc}] to identify which pairs were significantly different; Fig. 3a). The strongest significant linear correlations (Pearsons correlation) detected between dominant clade relative abundance and a broad range of environmental predictors had weak to moderate strength (*r* ≥ 0.3) with cloud- or wedge-shaped distributions (Fig. 3b).

**Microbial networks show shared supporting cohorts for the most abundant clades.** The full OTU network from both the time series and mixed cultures (see Fig. S1 in the supplemental material) shows modularity and taxonomic assortativity (grouping of related organisms or underdispersion) at all taxonomic levels with greater assortativity in the mixed cultures (see Table S1 in the supplemental material). Networks with such high modularity have dense connections between the nodes within modules but sparse connections between nodes of different modules. Moreover, assortativity marks a preference for a network's nodes to be linked to similar nodes. In our study, assortativity increased with taxonomic level peaking at phylum level in the time series, while assortativity was lowest at the phylum level in the cultures.

In a phylotype network, the first neighbors or primary cohort of an OTU may have preferential associations, or when negatively correlated, may be antagonistic or mutually exclusive. These patterns may for example be caused by metabolic dependencies or competition. In the network analysis of the lake time series and mixed cultures, the primary cohort (1° cohort) of "*Ca.* Planktophila," "*Ca.* Nanopelagicus," and "*Ca.* Fonsibacter" (Fig. 4a) were overlapping but to differing degrees. Specifically, 25% to 43% of the "*Ca.* Planktophila" 1° cohort were shared with "*Ca.* Nanopelagicus" and, although with a different set of phylotypes, basal "*Ca.* Fonsibacter" in the time series and mixed cultures. The "*Ca.* Fonsibacter" 1° cohort from the time series analysis overlapped only 10% with each of the other three cohorts. The OTUs shared between "*Ca.* Planktophila" and "*Ca.* Fonsibacter" had opposing correlation direction, while those shared between the two *Nanopelagicales* genera and between "*Ca.* Nanopelagicus" and "*Ca.* Fonsibacter" had the same correlation direction. Common to all three time series 1° cohorts were phylotypes of "*Candidatus* Nanopelagicus limnes" and *Polynucleobacter*. In the mixed cultures, half the "*Ca.* Fonsibacter" and "*Ca.* Nanopelagicus" 1° cohort OTUs were shared, while there was no unique overlap with "*Ca.* Planktophila" (Fig. 4b). Common to all three mixed cultures 1° cohorts were phylotypes of "*Ca.* Nanopelagicus limnes," *Flavobacterium*, and *Acinetobacter*.

The "*Ca.* Fonsibacter" 1° cohort network revealed a difference between the core group "*Ca.* Fonsibacter" and a phylotype representing a basal clade of this genus. The two groups were subsequently examined separately. The "*Ca.* Planktophila" 1° cohort also revealed a division between phylotypes of the "A6 tribe" and the others, as they had negative correlation to the other "*Ca.* Planktophila." This suggests that the "acI-A6 tribe" plays a different ecological role than the other clades in Lake Erken and/or may need to compete with the more dominant "*Ca.* Planktophila vernalis" of the acI-A7 clade, its closest relative.

There was a positive moderately strong correlation between the relative abundance of OTUs in the time series and relative abundance in cultures (Pearsons $r = 0.58$, adjusted $P$ value [$P_{adj}$] $= 0$; Table S2) but no other correlations between relative abundance or prevalence (percentage of samples where an OTU was detected).

**Some clades that were dominant in the lake were not similarly represented in cultures.** Both the actinobacterial "acIV" (C111) and *Comamonadaceae* families were more abundant in winter when the lake was covered by ice (Fig. 3a, KW $P < 0.001$, KWmc $P < 0.001$). While "acIV" was abundant and ubiquitous in the time series (Fig. 2b; range, 1% to 45%; av.ra, 11%), it was also prominent in the "*Ca.* Planktophila" time series 1° cohort (Fig. 4a). However, these phylotypes were not detected in the mixed cultures. *Comamonadaceae* were abundant but patchy in the time series and mixed cultures.

**Actinobacterial order *Nanopelagicales* (acI) was the most abundant clade in Lake Erken.** *Nanopelagicales* were the most abundant clade in Lake Erken (Fig. 2b) and were represented by 31 OTUs in the lake time series network with 21 "*Ca.* Planktophila," eight "*Ca.* Nanopelagicus," and one acI-c (Fig. S1). *Nanopelagicales* in Lake Erken had higher relative abundances in the hypolimnion (KW $P < 0.001$, KWmc $P < 0.001$) and were negatively correlated with percent $O_2$ (Pearsons $r = -0.4$, $P < 0.001$) (Fig. 3b).

**"*Ca.* Planktophila" was the dominant actinobacterial genus in the time series but was scarce in cultures.** Only 1 out of 21 "*Ca.* Planktophila" phylotypes was detected in the mixed cultures. The time series 1° cohort of "*Ca.* Planktophila" included other *Actinobacteria*, *Armatimonadetes*, *Chloroflexi*, *Gemmatimonadetes*, *Alpha-* and *Betaproteobacteria*, and *Verrucomicrobia*, while in the mixed cultures only *Actinobacteria*, *Bacteroidetes*, and *Alpha-* and *Betaproteobacteria* coexisted (Fig. S1). A pairwise comparison of the phylogenetic distance and correlation (averaged edge value from the correlation network) was conducted to see whether there was a relationship between OTU genetic relatedness and cooccurrence to distinguish between ecotype and community models. Within the "*Ca.* Planktophila" 1° cohort, there was no support for a correlation between phylogenetic distance and cooccurrence (Fig. 5), and "*Ca.* Planktophila" was the only genus tested that had negative intragenus correlations. "*Ca.*

## a) Lake Erken time-series

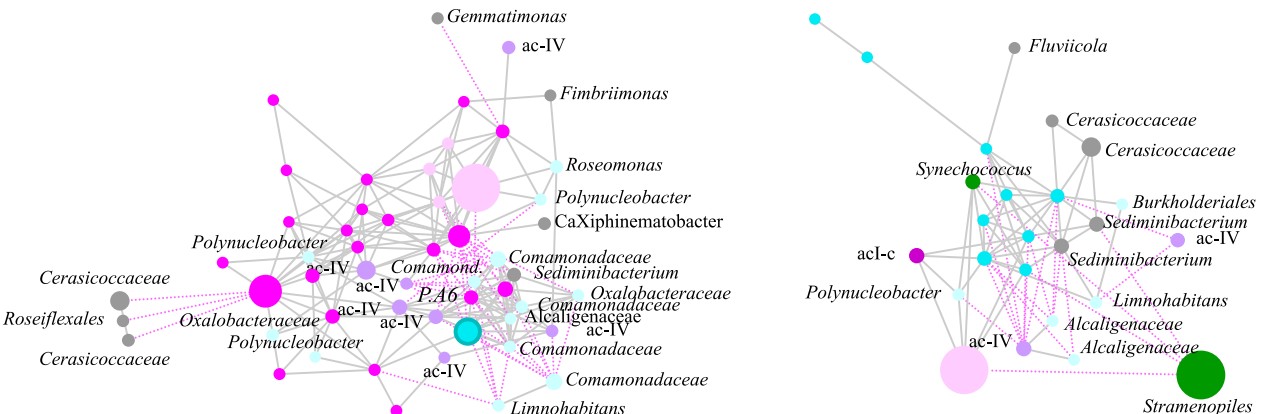

**"*Ca.* Planktophila" 1° cohort**

**"*Ca.* Fonsibacter" 1° cohort**

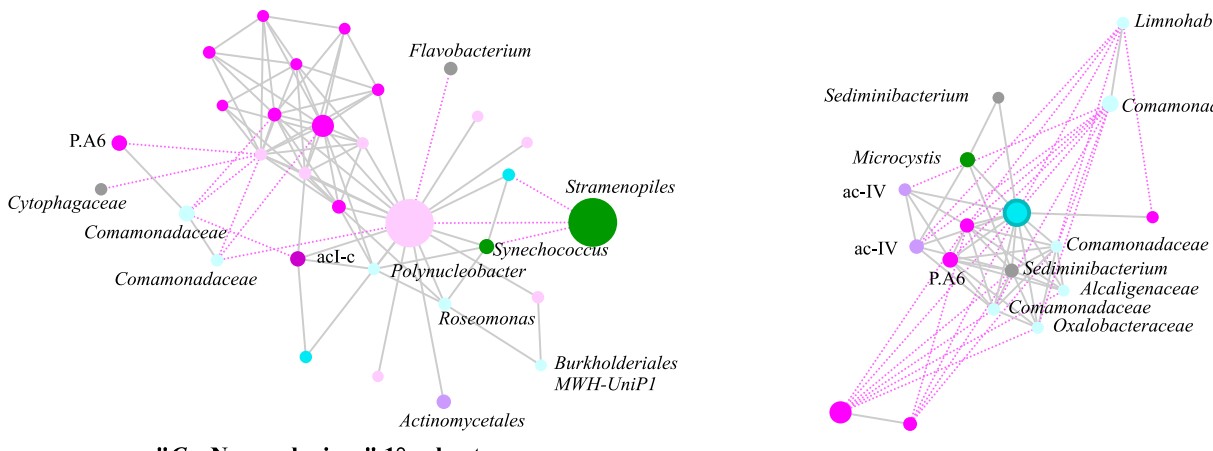

**"*Ca.* Nanopelagicus" 1° cohort**

**basal "Fonsibacter" 1° cohort**

## b) Combined 1° cohorts of mixed-cultures from lake water

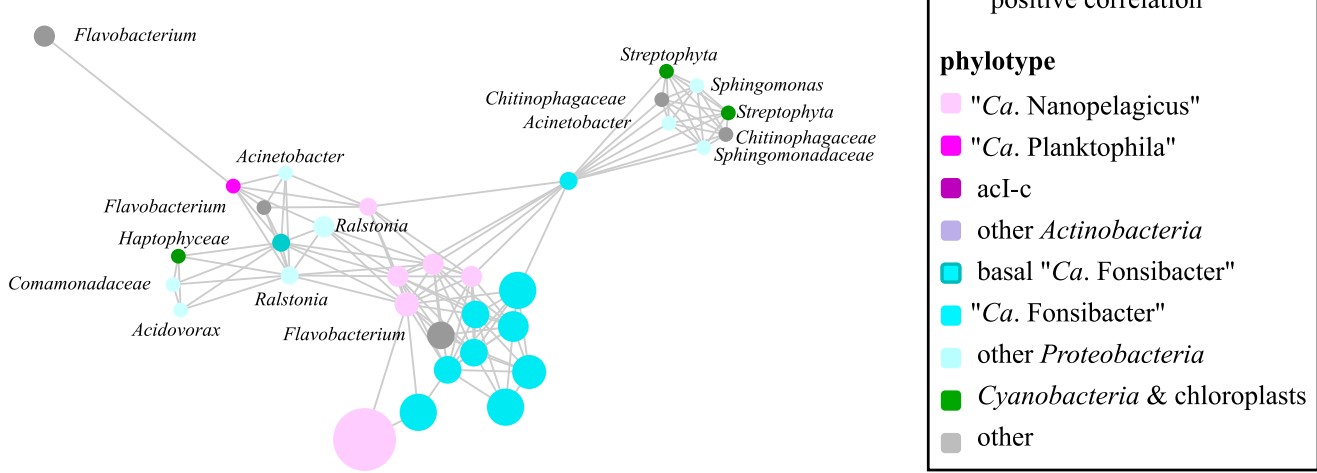

**edge type**

- ········· negative correlation
- —— positive correlation

**phylotype**

- "*Ca.* Nanopelagicus"
- "*Ca.* Planktophila"
- acI-c
- other *Actinobacteria*
- basal "*Ca.* Fonsibacter"
- "*Ca.* Fonsibacter"
- other *Proteobacteria*
- *Cyanobacteria* & chloroplasts
- other

**FIG 4** OTU networks of the 1° cohorts from amplicons of time-series separated genera (a) and mixed-cultures (b). Nodes sized by average relative abundance in time-series and by percentage of cultures detected in for mixed-cultures. The pink dotted lines indicate negative correlations between OTUs, and gray solid lines show positive correlations.

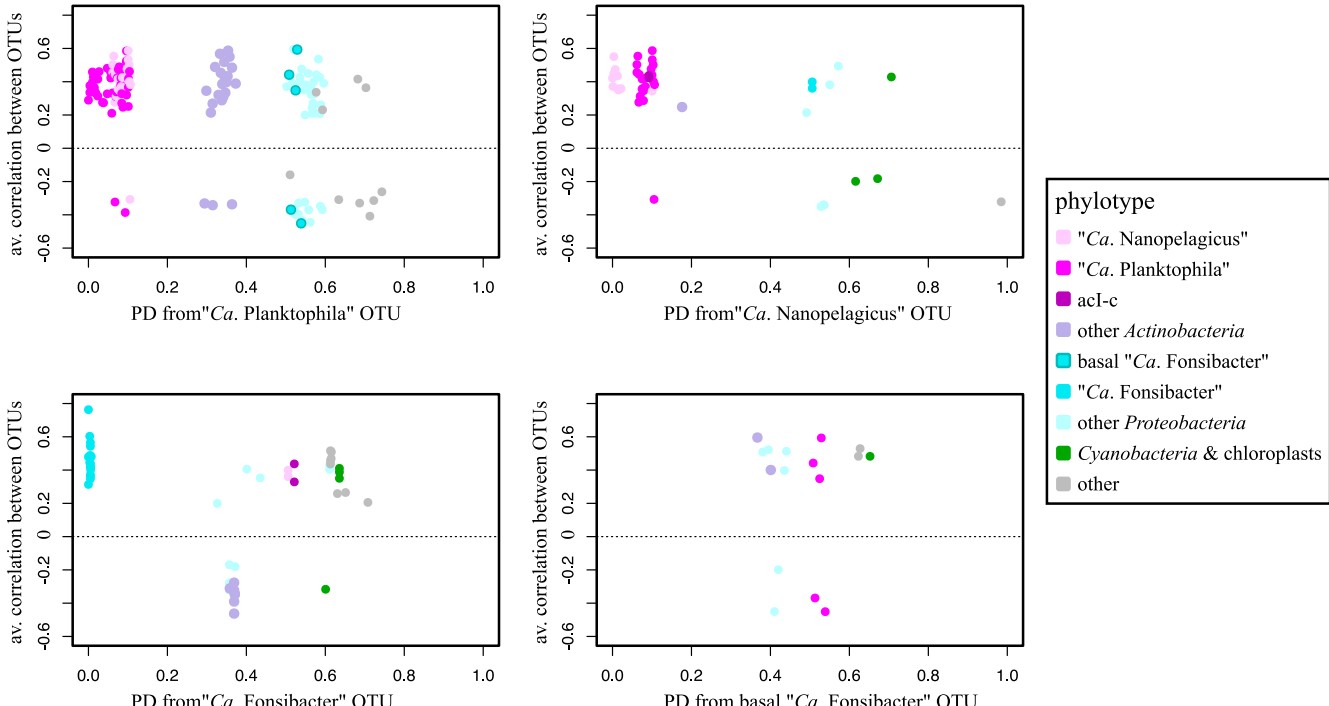

**FIG 5** Relationship between OTU phylogenetic distance (PD) and correlation of the 1° cohort phylotypes to selected genera: "*Ca.* Planktophila," "*Ca.* Nanopelagicus," "*Ca.* Fonsibacter," and basal "*Ca.* Fonsibacter."

Planktophila" had more intraphylum (i.e., non-*Nanopelagicales Actinobacteria*) correlations than the other genera tested. Unlike the other genera, "*Ca.* Planktophila" had no correlations (positive or negative) with phytoplankton in its 1° cohort (Fig. 4). Additionally, "*Ca.* Planktophila" was the dominant actinobacterial genus in the time series with "*Ca.* Planktophila vernalis" being most abundant (Fig. 4a). Conversely, in the mixed cultures, only one *Planktophila* phylotype was detected, "*Ca.* Planktophila limnetica" (Fig. 4b).

**"*Ca.* Nanopelagicus" was overall more prevalent in mixed cultures than in the lake time series.** Six out of nine "*Ca.* Nanopelagicus" phylotypes were detected in the mixed cultures. "*Ca.* Nanopelagicus" 1° cohort in the time series included other *Actinobacteria*, *Bacteroidetes*, *Alpha-* and *Betaproteobacteria*, and the phytoplanktonic *Synechococcus* and *Stramenopiles* (identified by chloroplast sequences), while in the mixed cultures there was a subset of these phyla with the addition of one gammaproteobacterial phylotype (Fig. 4). To calculate whether the presence of an OTU in the mixed cultures could be attributed to the probability of it being in the inoculum, the total and average relative abundances and the number of samples an OTU was detected in (prevalence) were compared across the two data sets with Pearsons correlation. There was no correlation between "*Ca.* Nanopelagicus" OTU relative abundance or prevalence in the lake time series compared to the mixed cultures (Table S2). "*Ca.* Nanopelagicus" was overall more prevalent in mixed cultures than in the lake time series. "*Ca.* Nanopelagicus" had a strong correlation with "Ca. Fonsibacter" in both data sets.

**The prevalence of alphaproteobacterial genus "*Ca.* Fonsibacter" in the lake time series strongly correlated with prevalence in cultures.** "*Ca.* Fonsibacter" spp. had higher relative abundance from winter to spring and moderate negative correlation with water temperature ($r = -0.4$, $P < 0.001$, Fig. 3b). "*Ca.* Fonsibacter" was represented by 10 OTUs in each of the time series and mixed-culture networks, with nine OTUs common to both and the alternating OTUs being the least abundant and least prevalent in both environments. The "*Ca.* Fonsibacter" 1° cohort from the time series includes other "*Ca.* Fonsibacter," *Bacteroidetes*, *Alpha-* and *Betaproteobacteria*,

*Actinobacteria*, including "*Ca*. Nanopelagicus" and acI-c, and phytoplankton *Synechococcus* and *Stramenopiles*. The "*Ca*. Fonsibacter" 1° cohort from mixed-culture experiments also included other "*Ca*. Fonsibacter" phylotypes, "*Ca*. Nanopelagicus," *Bacteroidetes*, other *Alpha-* and *Betaproteobacteria*, and macroalgal eukaryotes of the *Streptophyta* order. There was a strong correlation detected between "*Ca*. Fonsibacter" OTU prevalence in the lake time series compared to the prevalence of the respective OTU in the mixed cultures (corr = 0.82, $n = 11$, $P_{adj} = 0$, Table S2). Based on OTU identity, phylogenetic branch placement, and network connections, two "*Ca*. Fonsibacter" phylotypes were designated as basal and analyzed separately for 1° cohort. Unlike the core "*Ca*. Fonsibacter" phylotypes, these basal phylotypes had no negative correlations with phytoplankton and no correlations with other "*Ca*. Fonsibacter" phylotypes (Fig. 5).

**KatG, a growth factor for successful maintenance of axenic *Nanopelagicales* cultures.** After years of culture efforts by multiple groups, the successful maintenance of axenic *Nanopelagicales* cultures, "*Ca*. Planktophila rubra" and "*Ca*. Planktophila aquatilis" were finally achieved via the addition of an active catalase to culture media (25). Kim et al. (25) also sequenced the "*Ca*. Planktophila rubra" *katG* gene, cloned, expressed, purified, and assayed its catalase activity, finding that it was not a functional catalase. The presence or absence of *katG* and the homology and functionality of its product are therefore implicated in growth and culturability of the *Nanopelagicales*. Phylogenetic analysis of this gene showed that the *Nanopelagicales* feature complex presence/absence patterns of *katG* that separate the lineage into two groups inconsistent with only vertical transmission of the gene. From Lake Erken metagenomes, we recovered four "*Ca*. Nanopelagicus" metagenome-assembled genomes (MAGs) ranging from 18% to 72% completeness, 27 "*Ca*. Planktophila" MAGs between 15% and 85% complete, and three "*Ca*. Fonsibacter" MAGs with completeness between 18% and 27% (one basal "*Ca*. Fonsibacter" MAG having over 50% contamination). Analysis of (single) gene content and complementarities across such incomplete genomes was judged pointless. We therefore investigated the presence of the *katG* gene in all Lake Erken metagenome contigs to reduce false-negative results caused by insufficient MAG recovery. We also examined the *katG* gene neighborhood, including synteny among the *Nanopelagicales*, as well as the homology and topology of its translated product, KatG, with translated sequences from both public data sets and our metagenomes to look for evidence on how it may affect differences in culturability among the *Nanopelagicales*.

Phylogenetic analysis showed the same two groups as reported by Kim et al. (25). The "group A" which includes the previously studied "*Ca*. Planktophila rubra" KatG and other members of acI-A1 to A3 and "*Ca*. Nanopelagicus" (acI-B) MAGs, we designated the actinobacterial group (Fig. 6). Secondary and tertiary structure modeling of these KatGs found a consistent topology with a larger group that also included *Burkholderia*, *Neurospora*, *Haloarcula*, and *Mycobacterium* fold types but was closest to the KatG from *Burkholderia pseudomallei*. These actinobacterial KatGs had a predicted β-sheet around one of the active site residues instead of an α-helix and were 10 to 20 amino acid residues longer than the *B. pseudomallei* KatG. All known active site residues involved in heme interaction or enzyme switching were present. "*Ca*. Planktophila limnetica" (the only "*Ca*. Planktophila" phylotype detected in our mixed cultures) KatG had highest sequence identity with the assayed low-activity KatG from "*Ca*. Planktophila rubra" (IMCC 25003, acI-A1) which was located within a putative DNA repair region (Fig. 6 and Fig. S2). "*Ca*. Planktophila limnetica" (acI-A3), "*Ca*. Planktophila vernalis" (acI-A7), "*Ca*. Planktophila lacus" (acI-A4), and "*Ca*. Nanopelagicus limnes" (acI-B1) all have the repair cassette without *katG*. Lake Erken metagenomic actinobacterial group *katG* sequences were recovered matching "*Ca*. Planktophila dulcis," "*Ca*. Planktophila limnetica," and "*Ca*. Planktophila versatilis" (Fig. 6).

The second actinobacterial "group B" had the greatest sequence and topological homology with a larger group designated *Synechococcus* (Fig. 6) after the enzyme with the most information, and included "*Ca*. Planktophila lacus" (acI-A4), unnamed clades

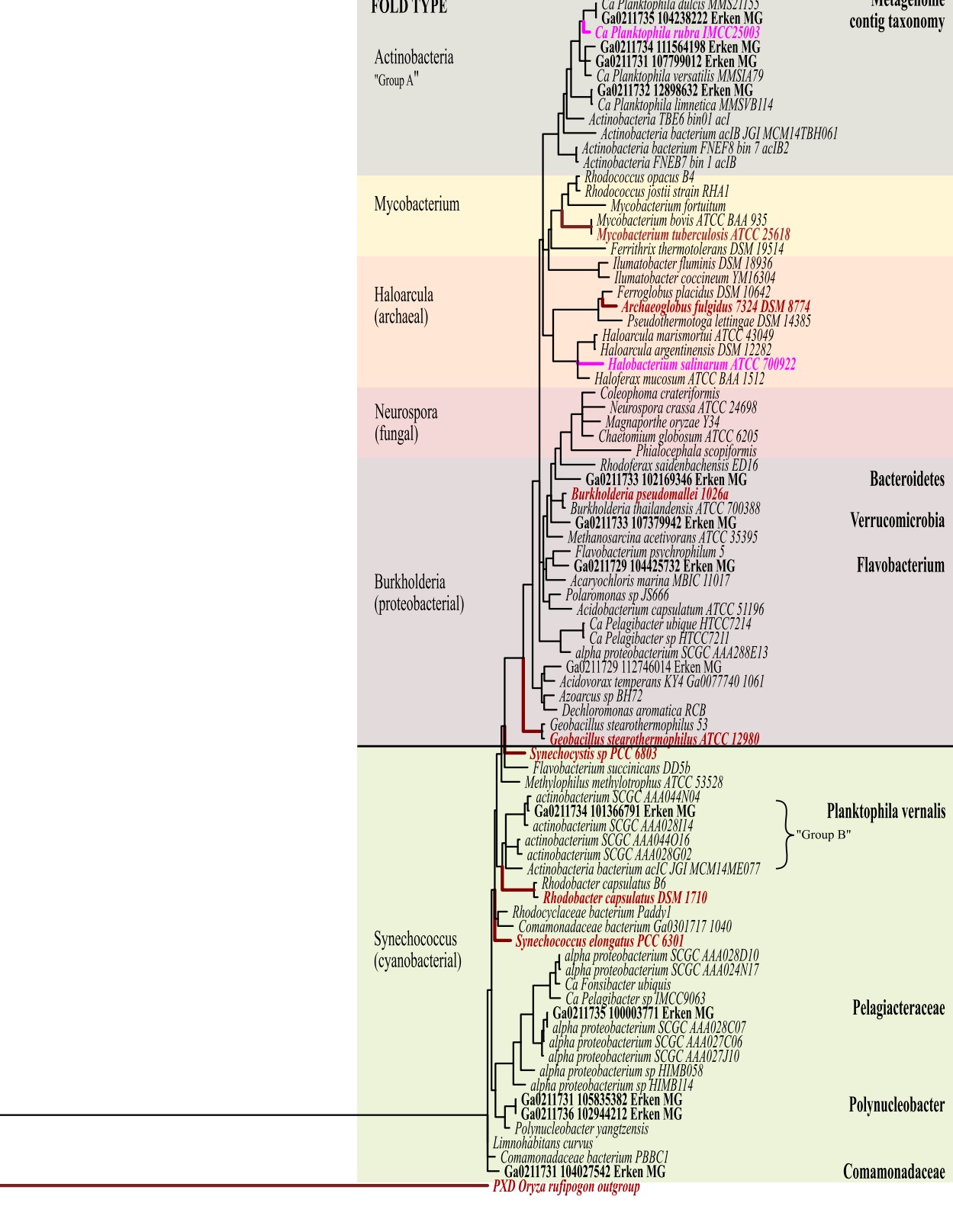

**FIG 6** RAXML phylogenetic tree of translated *katG* gene showing branches with >60% support. Bold taxon names are those with assay and/or crystallography data, high function in red and low function in pink. Lake Erken metagenomic contigs (names starting in GA) from *Nanopelgicales* and their 1° cohort are in bold black type with the assigned taxon listed at the far right.

acl-A5 and A6, and "*Ca.* Planktophila vernalis" (acl-A7). No anomalous folds or residues, at least compared to the highly functioning *Synechococcus* version of the gene, were detected by alignment or modeling. These *katG* genes were all found near a tRNA (Fig. S2). "*Ca.* Planktophila vernalis" and "*Ca.* Planktophila lacus" have the repair cassette seen in "group A" but without the *katG* gene.

The most abundant "*Ca.* Nanopelagicus" phylotypes in both data sets were assigned to the *katG* deficient "*Ca.* Nanopelagicus limnes" and no "*Ca.* Nanopelagicus" *katG* sequences were recovered from our metagenomes.

We also recovered metagenome *katG* sequences putatively matching "*Ca.* Fonsibacter" and the following 1° cohort members: *Gemmatimonas*, *Fimbriimonas*, *Bacteroidetes* (*Cytophacaceae*, *Fluviicoli*, *Flavobacterium*, *Chitinophagaceae*, *Sediminibacterium*, *Sphingobacteriales*), *Gammaproteobacteria* (*Acinetobacter*), *Verrucomicrobia* ("*Candidatus* Xiphinematobacter" and *Cerasicoccaceae*), *Betaproteobacteria* (*Polynucleobacter*, *Burkholderiales* MWH-UniP1, and *Comamonadaceae*) (Fig. 7).

The bifunctional catalase-peroxidase KatG has a heme ligand, and as such, the genetic, transcription, translation, and protein maintenance costs not only include the 700- to 800-amino-acid (aa)-long protein but also must include the cost for heme. Depending on whether glutamate (common in *Actinobacteria* [36]) or glycine (as seen in "*Ca.* Fonsibacter") is the base for the initial synthesis pathway to 5-aminolevulinate, an added cost will be either three genes (approximately 1,400 aa) or one gene (330 aa), respectively. From 5-aminolevulinate to protoheme and placement in KatG is a further seven steps by six or seven different gene products adding at least another 4,800 aa to the cost (36–38). The total cost therefore includes 8 to 11 genes with 5,800 aa to 7,200 aa in proteins. This is markedly expensive compared to the cost of a single gene encoding a (peroxi)redoxin of only 90 aa to 240 aa (38, 39). All nonphotosynthetic organism genomes of the three genera and their 1° cohort carried genes for peroxiredoxins which would be capable of mitigating intracellular reactive oxygen species (ROS) production at much lower metabolic cost than the production of KatG. Corrin rings as seen in heme are essential in heme-based catalase/peroxidases, cobalamin (vitamin B12), and chlorophyll. Except for the *Nanopelagicales*, all genomes analyzed here show heme biosynthesis capability if vitamin B12, photosynthetic, and/or KatG potential was also present (Fig. 7).

**Metabolic dependencies clarified interspecific relationships.** Metabolic dependencies can be overcome by scavenging biomolecules released during cellular death, and this type of relationship is more likely taking place between phylotypes that display negative correlations. Positive correlations can on the other hand be associated with cooperative cross-feeding and rely on partners being alive for an ongoing metabolic relationship. Many other relationships can be the cause of positive and negative correlations, but stable cross-feeding will not likely establish between organisms with opposing abundance profiles nor is broad scavenging of cellular detritus likely to occur between organisms with similar abundance profiles. "*Ca.* Planktophila" had positive correlations (Fig. 4) with 1° cohort phylotypes whose genomes encode the potential to meet all of its B-vitamin and heme dependencies (Fig. 7) with *Roseomonas* the only metagenomically confirmed encoder of vitamin B12 synthesis in the time series 1° cohort and *Ralstonia* in cultures. The predicted optional C and N source, cyanophycin, of "*Ca.* Planktophila" was likely *Polynucleobacter* and *Oxalobacteraceae* in the time series but *Flavobacterium* and/or *Ralstonia* in the mixed cultures. However, if cyanophycin is obtained as detritus, then the contenders are *Gemmatimonas*, *Limnohabitans*, *Alcaligenacaceae*, and *Comamonadaceae*.

The individual (meta)genomic contributions of the betaproteobacterial lineage designated Burkholderiales-MWH-UniP1 and *Comamonadaceae* phylotypes were difficult to separate, and it is probable that both can synthesize cyanophycin, as there were abundant and divergent sequences clustered in this lineage. Further, betaproteobacterial *Burkholderiales* class phylotypes (*Oxalobacteraceae*, *Limnohabitans*, *Comamonadaceae*, *Ral-*

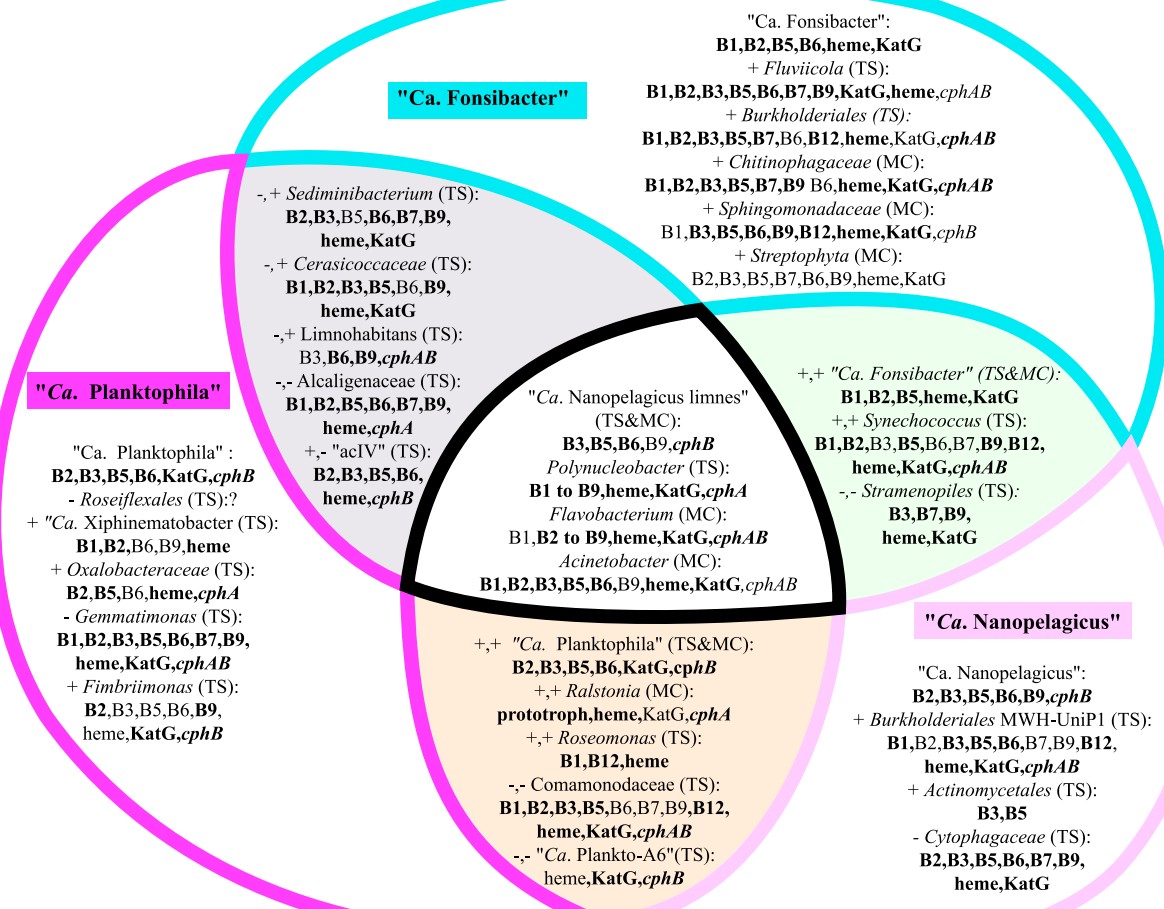

**FIG 7** Venn diagrams of the 1° cohorts with potential common goods encoded in genomes, genomic potential confirmed in lake metagenomes in bold. The plus or minus symbol in front of the genus indicates positive or negative correlation. All 1° cohorts were identified from time series (TS), both methods (TS&MC), or mixed cultures only (MC). The order of common goods listing is thiamine or vitamin B1, riboflavin or B2, niacin or B3, pantothenic acid or B5, pyridoxal-5P or B6, biotin or B7, folate or B9, cobalamin or B12, porphyrin ring or heme, bifunctional catalase/peroxidase or KatG, production of storage peptide cyanophycin or *cphA*, usage of cyanophycin or *cphB*. In this figure, "prototroph" refers to the coding potential to synthesize all B vitamins.

stonia, *Polynucleobacter*, *Alcaligenaceae*, MWH-UniP1, and unnamed) together likely contribute significantly to B-vitamin production in Lake Erken.

"*Ca*. Nanopelagicus" dependencies are similar to those of "*Ca*. Fonsibacter," except that the former could be a provider of common goods vitamins B3 and B9, while the latter could be a provider of vitamins B1 and B2, heme, and KatG. "*Ca*. Fonsibacter" dependencies are easily met by any of its positively correlated 1° cohort, though the strongest evidence suggests that all of its auxotrophies (except B12) could be met by the negatively correlated *Stramenopiles*. *Synechococcus* spp. are known to secrete B12 (40), and as there was an abundance of B12 synthesis sequences clustered for this clade in the metagenomes, they are the most likely provider of B12 to "*Ca*. Fonsibacter" even if *Sphingomonadaceae* and *Burkholderiales* also had metagenomically confirmed potential.

## DISCUSSION

**Dominance of the abundant streamlined genera is linked to their interactions.** If microbial abundance is directly linked to nutrient availability or other physicochemical parameters, then a strong correlation between geochemical parameters and relative abundance is expected. The lack of strong correlation between OTUs or clades and

environmental parameters in Lake Erken in Sweden indicates that during the 8 years of sample collection, microbial community composition was not directly and strongly controlled but only weakly influenced by the measured abiotic environmental variables. While short-term studies have shown significant strong trends and correlations (12), such correlations tend to be eclipsed in studies spanning multiple years by shifts operating at longer frequencies and persistent biotic interactions (41–44). Overall, the strength and number (identified via consensus network) of correlations between taxon relative abundance and the weakness of environmental predictors for most of the taxa suggest that biological aspects (including predation, which we did not study) are the greater determinants of microbial population dynamics, composition, and persistence in the studied lake. It stands therefore that the dominance of the three streamlined genera in the lake is attributable more to their interactions with their community than selection directly controlled by the environmental state.

**Growth of bacteria in dilution mixed cultures as a predictor of ecological strategy.** Assortativity is a predicted product of ecotype or niche differentiation at the species level and environmental filtering at higher taxonomic levels (45). Ecotype divergence of strains reduces competition between different members of a population. The detection of the highest assortativity in the time series networks at the phylum level confirms environmental selection as important in the lake habitat but not at the more highly resolved taxonomic levels. It is unlikely that environmental filtering is the main driver behind the success of these three genera in Lake Erken. High assortativity in the mixed cultures at all taxonomic levels except phylum suggests that niche differentiation contributed to cooccurrence in the cultures, though other factors, including ecotype cooperation, may have contributed. Neither result however fully supports, or refutes, ecotype theory as the explanation of coexistence or dominance by "*Ca.* Planktophila," "*Ca.* Nanopelagicus," and "*Ca.* Fonsibacter" genera.

The strong correlation between "*Ca.* Fonsibacter" (and also *Comamonadaceae*) prevalence in the time series and mixed cultures supports that its occurrence in the cultures may be attributed to being present in the inocula and that the filtered lake water media were able to meet an important proportion of its metabolic needs. Conversely, the lack of correlation in either relative abundance or prevalence for the *Nanopelagicales* genera (and also the "acIV" family) in the time series compared to cultures means that there is not support for their detection being merely due to high abundance in the inocula. We infer that the other phylotypes were bigger contributors to their growth. It should be noted that negative correlations are not detectable in the mixed cultures, as the absence of a phylotype could be stochastic rather than a result of competitive exclusion. Moreover, competition that prevents growth in a given culture cannot be identified.

***Nanopelagicales* in Lake Erken likely use a combination of community cooperation and detritus ecological strategies.** *Nanopelagicales* in Lake Erken were likely more competitive in colder, darker, and oxygen-depleted waters. While *Nanopelagicales* have been associated with oxygen-saturated waters in some locations (46), this lineage has been associated with colder deeper water at other locations (30). This variation in association could be due to the considerable genetic diversity encompassed in this bacterial order (47), and future research would benefit from examining its "species" separately. It is also probable that solar-driven formation of oxygen radicals in surface waters inhibits the growth of *katG* deficient species of *Nanopelagicales*.

Of the seven *Nanopelagicales* phylotypes in the mixed cultures, six were members of the candidate genus "*Ca.* Nanopelagicus," suggesting that dependencies of this genus were more easily met compared to the "*Ca.* Planktophila" spp. found in Lake Erken. However, genomic evidence from cultures from Lake Zurich has shown the opposite, as "*Ca.* Nanopelagicus" spp. have more auxotrophies than "*Ca.* Planktophila" spp. (23). It is therefore likely that the difference in culturability seen in this experiment was due to reduced costs via gene loss, i.e., the Black Queen hypothesis, in combination with suitable metabolic partners being in the inoculum or different auxotrophies of the clades in Lake Erken. Specifically, the lower maintenance costs due to the loss of *katG*

(25, 33) and amino acid synthesis could increase fitness of the "*Ca*. Nanopelagicus." The successful maintenance of "*Ca*. Planktophila" spp. in culture pinpointed the low functionality of catalase-peroxidases as the culprit for the difficulties in growing these bacteria. The most parsimonious explanation for *Nanopelagicales katG* presence/absence/homology is that the now low-functioning version is in the process of being lost and that the "*Ca*. Planktophila" spp. with a high-functioning KatG obtained this version via a horizontal gene transfer (HGT) event. The retention of a low-functioning KatG could be due to the criticality of the repair cassette whereby only lineages where *katG* is excised without disturbing the surrounding region will survive, or to the other (putative) functions of KatG, including as a detoxifier of formate and methanol. Future research assaying "actinobacterial group" KatGs for activity with other low-molecular-weight (LMW) molecules such as formate would be informative. While "*Ca*. Planktophila vernalis" in Lake Erken is likely *katG*$^+$, questions remain as to the functionality of even these KatGs, as no heme synthesis pathways have been recorded for any *Nanopelagicales*, and without the heme ligand, there would be no catalytic function. Future work assaying "group B" KatGs and also showing evidence of hemophore secretion or membrane heme receptor binding would clarify this issue. It is likely that "*Ca*. Planktophila" and "*Ca*. Nanopelagicus limnes" from Lake Erken rely on *Fimbriimonas*, *Polynucleobacter*, *Synechococcus*, and "*Ca*. Fonsibacter," while "*Ca*. Planktophila limnetica" and "*Ca*. Nanopelagicus limnes" in culture relied on a combination of *Flavobacterium and Acinetobacter* to detoxify environmental peroxide. This fits within the community cooperation model.

The higher culturability of "*Ca*. Nanopelagicus" suggests that it may partially be due to the cost of harboring *katG*, whether high or low functioning, because *katG* is larger than average (48), and cells can rely on other community members for oxidative stress reduction from sunlight- and phototroph-generated ROS. Further, the greater reliance on exogenous sources of amino acids in "*Ca*. Nanopelagicus" fits within the Black Queen hypothesis (BQH) and the community detritus model. A discussion on where to draw the theoretical line for (potential) BQH at a single gene (e.g., KatG), interconnected pathway (e.g., heme plus KatG), or associated systems (e.g., photosynthesis, respiratory electron transport, heme, KatG, etc.) could be fruitful in light of the ubiquity and abundance of streamlined genomes in aquatic environments.

**"*Ca*. Fonsibacter" uses mostly detritus to meet its dependencies but relies on ecotype differentiation for defense.** Earlier short-term studies found that "*Ca*. Fonsibacter" is highly abundant in Lake Erken in the late summer-early autumn immediately after an algal bloom, i.e., the "clear water" phase of the lake's cycle (12). However, we found support for higher long-term relative abundances in winter and spring and an associated negative correlation with water temperature. These different findings could be due to changes in abundance between years of different ecotypes (with different temperature and nutrient preferences) that were not detected via 16S rRNA amplicon surveys. The ecotype differentiation within "*Ca*. Fonsibacter" is centered around the HVR1 which is thought to encode genes for defense, i.e., predator escape via alteration of potential receptor sites (viral infection) and cellular membrane qualities (grazing) (3, 49). There appears to be limited intrapopulation variability in anabolic or catabolic traits (14, 24, 50), and as such, ecotype differentiation for overcoming auxotrophies is not supported. While no direct evidence exists for "*Ca*. Fonsibacter," its similarity to the marine sister clade Pelagibacter suggests that it has overcome auxotrophic limitations by scavenging metabolites and other compounds produced by phototrophs (51). In Lake Erken, this would be primarily dying and senescent *Stramenopiles* as reflected in the apparent negative correlation. Evidence points to "*Ca*. Fonsibacter" relying on detritus to meet most of its dependencies and ecotype differentiation for defense. Vitamin B12 was the only organic molecule identified here that could require cooperation, as evidenced by positive correlation which implies secretion rather than cell death, with potential partners being *Sphingomonadaceae* and *Synechococcus*. Culture experiments comparing "*Ca*. Fonsibacter" culture using supernatant from po-

tential dependency providers (*Sphingomonadaceae*, *Synechococcus*, *Stramenopiles*) versus coculture would clarify their relationships.

The combination of network and other correlation analyses with review of published genomes and metagenome dependency confirmation supports that most *Nanopelagicales* have anabolic community cooperation and community detritus with reliance on detoxifying community cooperation as the main structuring factors. "*Ca*. Fonsibacter" likely escape predation via ecotype cooperation, while broadly relying on community detritus for anabolic deficits.

## MATERIALS AND METHODS

**Time series water sampling and environmental parameters.** Time series water samples were obtained from dimictic Lake Erken (59.1510N, 18.1360E) in central Sweden at monthly to weekly intervals spanning a period from February 2007 to November 2015. Samples for genomic and chemical analyses were taken from the 20-m water column at 1-m intervals from the deepest point of the lake (Fig. 2a). Samples for the total water column were pooled during periods of mixing. Samples of the upper oxygenated water column (epilimnion), middle water column (metalimnion), and lower oxygen-depleted water column (hypolimnion) were pooled into separate composite samples during summer stratification. Annual dates of mixing, stratification, ice over, and ice thaw varied, as did depth of stratigraphic layers. Temperature and oxygen concentrations were analyzed every meter with a portable Oxi 340i oxygen meter equipped with a Cellox 325-20WTW probe and were used to determine stratification. Chemical analysis of pooled water samples was carried out by the Lake Erken Field Station International Organization for Standardization (ISO)-certified laboratory and included pH, conductivity, nitrate, nitrite, alkalinity, turbidity, suspended matter, phosphate, ammonium, total and particulate phosphorus and nitrogen, chlorophyll *a*, suspended particulate organic matter, silicate, water color, and absorbance at 420 nm measurements. Water samples for genomic analysis were collected by gentle vacuum filtration onto 0.2-$\mu$m membrane filters (Supor-200 membrane disc filters, 47 mm; Pall Corporation, East Hills, NY, USA). Filters were individually stored in Eppendorf tubes and transported back to the Uppsala University laboratory where they were stored at $-80°$C until further processing.

**DNA extraction and SSU rRNA gene amplicon preparation.** DNA was extracted from filters using MoBio UltraClean soil DNA extraction kits (MoBio, Carlsbad, CA, USA) per the manufacturer's instructions. The V3-V4 region of the 16S rRNA gene was amplified using the S-d-Bact-0341-b-S-17 (Bakt_341F, CCTACGGGNGGCWGCAG) forward primer and S-d-Bact-0785-b-A-21 (805RN, GACTACNVGGGTATCTAATCC) reverse primer (52). Template DNA was amplified in duplicate 20-$\mu$l reaction mixtures containing 1.0 U of Q5 high-fidelity DNA polymerase (NEB, UK), 0.25 $\mu$M primers, 200 $\mu$M deoxynucleoside triphosphate (dNTP) mix, and 0.4 $\mu$g bovine serum albumin (BSA). The thermocycler program was an initial denaturation step at 95°C for 30 s, followed by 20 cycles, with 1 cycle consisting of dissociation at 95°C for 10 s, annealing at 53°C for 30 s, extension at 72°C for 30 s, with a final extension step of 2 min at 72°C. Amplicons were pooled and purified with Agencourt AMPure XP purification system (Beckman Coulter, Danvers, MA, USA). Purified amplicons (2 $\mu$l) were amplified in second step 20-$\mu$l reactions to introduce sample multiplex identifiers (MIDs) to each end of the amplicons. Reactions contained 1.0 U Q5 high-fidelity DNA polymerase (NEB, UK), 0.25 $\mu$M primers, 200 $\mu$M dNTP mix, and 0.4 $\mu$g BSA. The thermocycler program was an initial denaturation step at 95°C for 30 s, followed by 15 cycles, with 1 cycle consisting of dissociation at 95°C for 10 s, annealing at 66°C for 30 s, and extension at 72°C for 30 s, with a final extension step of 2 min at 72°C. Amplicons were again purified using Agencourt AMPure XP purification system (Beckman Coulter, Danvers, MA, USA) and then quantified using PicoGreen (Invitrogen) before pooling at equimolar amounts for each run. Amplicons were sequenced at SciLifeLab SNP/SEQ service on the MiSeq Illumina platform with 300-bp paired-end libraries. MiSeq data were processed, including demultiplexing, by the sequencing center using Illumina pipelines where all reads with more than 8% mismatch to adapter-MID sequences were discarded.

**Water sampling, culture conditions, and DNA amplification for mixed cultures.** Water as both a source of growth media and culturable cells was collected from the epilimnion of Lake Erken on 8 March 2016 from the marked sample point (Fig. 2a). Dilution mixed cultures were prepared according to the schematic shown in Fig. 2b but described in detail here. Water for growth media was filter sterilized twice through Sterivex filter cartridges (0.22 $\mu$m) (Millipore). Ten-minute exposure to UV light further disinfected and disrupted cellular and viral nucleic acids. Ten microliters of untreated lake water ($10^6$ cells/ml) was used as a source of cells for inoculation of mixed cultures. This untreated lake water was diluted in 1 liter of the triple sterilized lake water (approximately 10 cells/ml). Ninety-eight 1-ml cultures were placed in sterile 50-ml Falcon tubes to allow oxygenation and incubated for 2 months under 12-h light-dark cycles at 11°C to replicate the lake conditions at the time of collection. The V3-V4 region of the 16S rRNA gene was amplified using the S-d-Bact-0341-b-S-17 (Bakt_341F, CCTACGGGNGGCWGCAG) forward primer and the S-d-Bact-0785-b-A-21 (805RN, GACTACNVGGGTATCTAATCC) reverse primer (52). One microliter of the mixed culture was, after one freeze-thaw cycle, used as the PCR template, for Q5 High-Fidelity DNA polymerase. The PCR conditions consisted of an initial step at 98°C for 10 min. Thereafter, 35 cycles were run, with 1 cycle consisting of denaturation at 98°C for 10 s, annealing for 30 s, and extension at 72°C for 30 s. The annealing temperature was 48°C which has been tested to obtain unbiased products of nonmismatch and three-mismatch isolates. The final extension step was performed at 72°C for 2 min. PCR products were purified with magnetic beads (Angecourt AMPure). A second PCR

was conducted for attaching standard Illumina handles and MIDs as described above. Only data from the 60 cultures with a visible band after the first PCR were included in analyses.

**Amplicon bioinformatics.** Demultiplexed MiSeq time series and mixed-culture data, and 454 Ti time series data obtained from Lake Erken in 2008 (11), were preprocessed separately and joined once chimeras had been removed. Preprocessing of MiSeq data started with SeqPrep v1.2 (https://github .com/jstjohn/SeqPrep.git) (53) with default settings to join paired-end reads while discarding reads shorter than 30 bases and all unpaired reads. 454 and MiSeq reads with ambiguous bases, low-quality average, and low-quality sections, were discarded and trimmed with Qiime v1.9 (54) split_libraries and multiple split libraries fastq, respectively. Singletons and sequences identified as chimeric were removed with Qiimes' identify_chimeric_seqs, filter_fasta using usearch v6.1 (55), with default settings and option parameter "intersection" to select out sequences identified by both *de novo* and reference-based methods. Both GreenGenes (v105 13-8) and SILVA (v128) reference data sets were used (56, 57). Sequences from the two MiSeq runs and the 454 data from 2008 were then combined. Reads were clustered at 97% identity into OTUs and then assigned taxonomy based on SILVA v128 using QIIMES closed pick OTU method (uses only template recognition and does not allow for *de novo* sequences) to allow for differences in DNA processing and sequencing across the three data sets. Sequences from the negative control were removed from the entire data set. Samples with potential human microbiome contamination were identified and discarded from the data set due to potential effect on community metabolism. Sequences identified as eukaryotic (including fungal, mitochondrial, and chloroplast origin) or archaeal were deliberately retained despite the primers not specifically being selected to amplify such sequences, as they were considered critical in the mixed-culture analyses from a community metabolism perspective. Samples were normalized using Qiimes single rarefaction to an even depth of 1,500 reads per sample for the time series and 500 reads for the mixed-culture samples. These normalized OTU tables were then used for all downstream analyses.

**Statistical analyses of time series environmental correlates and selected taxa.** Statistical analyses were done using R v3.3.1 (58) in the RStudio IDE v0.99.903 (59), and graphs were processed for publication in Inkscape v0.91. Environmental parameters were removed if more than 40% of the values were missing, then the remaining 21 were tested for colinearity with the usdm v1.1.18 package with vifstep followed by vifcor functions and removed if the variance inflation factor (VIF) was >10 or correlations were over 0.7 sequentially starting with those with the highest number of missing values. The remaining 14 environmental predictors were tested for statistical differences in connection to the lake cycle with Kruskal-Wallis (kruskal.test of the base stats package) and graphed using beanplot v1.2 (60) and scales v0.4.1. Statistical significance of differences between individual paired cycle points were determined *post hoc* with the Kruskal-Wallis multiple comparison (KWmc) using the kruskal.mc test of the pgirmess package v1.6.4 (61). Environmental parameter correlates with selected taxon relative abundances were screened using Pearsons correlation with an alpha level of 0.001, and $P$ values were estimated and corrected for multiple comparisons with Benjamini, Hochberg, and Yekutieli (BH) false discovery rate using the psyche package v1.6.12 corr.test function and Pearsons moment $r$ reported. Parameters with weak to moderate correlation ($r > |0.3|$) were plotted in R to observe monotonicity.

**Network analyses.** The normalized time series OTU table was further processed for network and correlation analyses to remove rare OTUs that, while potentially important within the community, are known to inhibit identification of ecological relationships between OTUs and also between OTUs and their environment. Recommendations include OTU tables with 50% or less sparsity and $n_{eff}$ (inverse Simpsons) above an average of 10 and the use of multiple methods (62). A range of filtering parameters based on frequency of OTU detection, from 2 samples up to 46 (20% of samples) and total relative abundance ranging from 1 up to 150 (0.1%), was used on the OTU table to remove type I errors in detection of cooccurrence networks (62). The best filter combination was found with OTUs at a total relative abundance above 0.1% and occurring in at least 29 samples. All OTU tables were analyzed in R (58) using the vegan v2.4-1 package (63) diversity function to find the $n_{eff}$ and the base stats v3.3.1 package to find the minimum, maximum, and mean of the $n_{eff}$. The reduced OTU table was analyzed with SparCC (64), Pearsons correlation coefficient, and SPIEC-EASI (65). SparCC analyses were carried out in Rstudio (59) using both the rsparcc and spiec.easi packages with rsparcc parameters max.iter = 100, th = 0.1, and exiter = 10 and pseudo $P$ values generated with spiec.easis sparccboot (100 bootstraps) and pval.sparccboot scripts (two-sided test). Pearson correlations were performed using the corr.test of the psyche package using the optional $P$ values adjustment for multiple testing of Benjamini, Hochberg, and Yekutieli false discovery rate. SPIEC-EASI correlations were analyzed using the spiec.easi package with parameters method = "mb," sel.criterion = "stars," lambda.min.ratio = 1e−2, nlambda = 20, and 50 repetitions. Singletons were removed from the normalized mixed-culture OTU table prior to network analysis. The reduced OTU table was analyzed with Pearsons correlation, SpiecEasi (65) and on the binary version of the table, Dice-Sørensen index (66, 67).

Correlations between OTUs in the time series and mixed cultures were retained if the correlation coefficient was greater than 0.3 absolute for Pearsons, SparCC, or Dice-Sørensen or if detected at all by SPIEC-EASI. Time series and culture networks consisting of OTU connections detected by at least two correlation methods were visualized in Cytoscape. All further statistical analyses of the networks were performed in Cytoscape except for assortativity analyses which were done in R using the igraph v1.1.2 package nominal assortativity function. Subnetworks of the identified dominant genera ("*Ca.* Plankto-phila," "*Ca.* Nanopelagicus," and "*Ca.* Fonsibacter") and their 1° cohorts were extracted from the two initial networks and visualized separately. The layouts for the networks were "edge-weighted spring-embedded" with edge weights calculated from the average of the correlations. Some nodes (OTUs) were moved to the side of overlapping nodes for clarity or away from an edge if it appeared to pass through

an unconnected node. Correlations between *Nanopelagicales* OTU and "*Ca*. Fonsibacter" OTU detection in the time series versus the mixed cultures were investigated by Pearson correlation of their relative abundance (log av.ra), prevalence (percent samples or cultures detected in), and relative abundance versus prevalence to determine whether detection in culture was correlated with occurrence in the lake.

**Metagenomic sequencing.** Samples for metagenomic sequencing were selected by highest relative abundance of clades of interest according to amplicon data, greatest available DNA mass, and greatest variation in environmental parameters. This was done to examine samples with the greatest potential divergence of environmental conditions under which the microbes of interest were most abundant. These categories covered four lake cycle points, one from lake mixing and one each from the hypo-, meta-, and epi-limnionic layers during summer stratification. Two samples per category were selected from different years. Lower biomass during ice cover meant there was insufficient DNA for metagenomic sequencing. DNA library for metagenomic sequencing was prepped using the Illumina TruSeq Nano and 550-bp NeoPrep and sequenced with Illumina HiSeq 2500 high-output V4 paired-end (PE) 2x125bp at SciLifeLab Stockholm.

**Metagenome bioinformatics.** HiSeq Illumina paired-end reads were preprocessed first using SeqPrep v1.2 (53) to filter out low-quality reads using default settings resulting in removal of all reads less than 30 bases in length, unpaired reads, and initial clipping of adapter sequences. Next, the sequences were cleaned using Trimmomatic v0.36 (68) with removal of any remaining Illumina adapters allowing for two mismatches, followed by 4-base sliding window trim with a minimum average quality of 15, and then trimming of read ends with "maxinfo" value of less than 1. The eight metagenomes were individually assembled with megahit (69) using iterative kmers from 21 to 121 at 10-base intervals, minimum coverage of 2, and minimum length of 200. Concatenated assemblies were cleaned of Illumina artifact low-complexity (long homopolymer) contigs with prinseq-lite v0.20.4 (70), removing contigs consisting of greater than 80% one nucleotide. Mixed-assembled reads were then mapped back to their original metagenomes using bowtie2 v2.2.6 (71), and sam files were converted to bam files and sorted using samtools v1.5 (72) and deduplicated with picard v2.10.3 (https://github.com/broadinstitute/picard .git). Mixed-assembly contigs were binned into putative MAGs with metabat2 v2.12.1 (73), and bins were validated with CheckM (74), hmmer v3.1b2 (75), and pplacer v1.1 alpha19 (76). The taxonomy of MAGs were checked by placement within a larger tree using PhyloPhlAn (77), and the completeness and contamination estimates of MAGs that matched 1° cohort phylotypes were noted.

**Investigation of *katG* gene and its product, catalase-peroxidase I.** Known impediments to axenic growth of freshwater isolates are multiple auxotrophies and slow growth and replication rates. The successful axenic cultivation of "*Ca*. Planktophila" strains by the addition of high-functioning catalase to reduce oxidative stress, in addition to catering for predicted auxotrophies, showed that there can be other considerations. To investigate how both "*Ca*. Planktophila" and "*Ca*. Nanopelagicus" phylotypes grow in Lake Erken and why "*Ca*. Nanopelagicus" grew more frequently in the minimalist mixed cultures, the *katG* gene was investigated. First, KatG amino acid sequences for species identified as belonging to the *Nanopelagicales* and its 1° cohort were retrieved from IMG and GenBank (38, 78). Second, sequences for whom the gene product has been assayed and/or the three-dimensional (3D) crystal structure is known were identified via Phyre2 and then collected from RCSB PDB (rscb.org [79, 80]). Next, these sequences were used as BLAST (81) queries to retrieve closely related (>80% amino acid identity) sequences from Lake Erken metagenomes. The collected amino acid sequences were aligned using MAFFT e-insi and "leavegappy" options (82). Sequences with greater than 99% identity were identified using CD-Hit and removed (83), while sequences from uniRef useful for constructing a guide tree were then also added (84, 85). The alignment was evaluated for best amino acid substitution model using IQ-tree v.1.6.12 modelfinder (86). This alignment was then analyzed and used to construct a tree in RAXML v.0.9.0 (87) with WAG+IG+R4 model and 1,000 bootstraps, and catalase-peroxidase from *Oryza rufipogon* was set as an outgroup. To examine functionality, secondary structure was predicted and aligned to known models, and then tertiary structure was predicted in Phyre² (80). Further, the locations of *katG* genes in *Nanopelagicales* genomes and MAGs were examined in IMG using the gene neighborhood function. Images were downloaded with screen shot and processed in Inkscape.

**Other metabolic dependencies as found encoded in 1° cohort genomes and Lake Erken metagenomes.** To investigate other possible metabolic dependencies among the three genera and their 1° cohorts, we catalogued, based on public genome annotations, the probable B group vitamin interactions and the putative C and N source utilized by *Nanopelagicales*, cyanophycin. Taxa representative of either a group or closest individual genome to 1° cohort phylotypes were collated via the following NCBI taxon identifiers (IDs): *Cerasicoccaceae* txid2026784, *Chitinophagaceae* txid563835, *Comamonadaceae* txid80864, *Comamonadaceae Acidovorax* txid12916, *Cytophagaceae* txid2026784, *Fimbriimonas* txid661478, *Flavobacterium* txid237, *Fluviicola* txid2597671, *Gemmatimonas* txid1379270, *Limnohabitans* txid665874, *Oxalobacteraceae* (B-prot-Burkh) txid75682, *Polynucleobacter* txid556054, *Polynucleobacter acidiphobus* txid556053, *Ralstonia* txid48736, *Roseomonas* txid125216, *Sediminibacterium* txid1086393, *Sphingomonadaceae* txid41297, *Stramenopiles* (diatom) txid296543, *Synechococcus* txid1129 and txid32046, and "*Ca*. Xiphinematobacter" txid1704307. For actinobacterial genomes available only from IMG, the taxa were aclV acAcidi 2654587759 and 2739367537 and 2737471702 and 2737471663, actinobacterium acI-A6 SCGC AAA028-I14 2619618809, *Actinomycetales actinobacterium* SCGC AAA027-F04 2606217193, and actinobacterium SCGC AAA028-N15 2619618810. No genetic information was available for "Streptophyta" so a text-based reference search was conducted. No genetic information was available from the proposed *Chloroflexi* class *Roseiflexales*. Genes considered indicative or essential in a synthesis pathway were examined by search word against the taxon ID in NCBI or if in IMG then via KEGG pathway tools and BLAST. Pathways with NCBI search terms in parentheses and EC

numbers used to retrieve sequences from UniProt UniRef database (84) were: *cphAB* "cyanophycin*" EC 6.3.2.29/6.6.2.30 and EC 3.4.15.6, *katG* "catalase*" EC 1.11.1.21, ROS-"catalase*" and "peroxiredoxin," thiamine (B1) "thiazole synthase" EC 2.8.1.10, riboflavin (B2) "6,7-dimethyl-8-ribityllumazine synthase" EC 2.5.1.78, niacin (B3) "nicotinate-nucleotide diphosphorylase" EC 2.4.2.19, pantothenic acid (B5) "pantoate beta-alanine ligase" EC 6.3.2.1, pyridoxal (B6) synthesis "Pyridoxamine 5'-phosphate oxidase" EC 1.4.3.5, B6 salvage "Pyridoxamine kinase" EC 2.7.1.35, biotin (B7) "Dethiobiotin synthase" EC 6.3.3.3 and "8-amino-7-oxononanoate synthase" EC 2.3.1.47, folate (B9) "Dihydrofolate reductase" EC 1.5.1.3, cobalamin (B12) "*cobyric acid synthase" EC 6.3.5.10 cobQ/cbiP, "precorrin-6A" CobF EC 2.1.1.152, and "Cobalt-precorrin-5B C(1)-methyltransferase" CbiD EC 2.1.1.195, porphyrin (heme) "Porphobilinogen deaminase" and "hydroxymethylbilane synthase" EC 2.5.1.61. Representative sequences from bacterial and archaeal domains were obtained from UniProt nr90 for each gene using the listed EC numbers, and short fragments and metagenome sequences were removed. Annotated sequences from the Lake Erken metagenomes were downloaded from IMG using the KEGG pathway function and clustered at 95%, and sequences shorter than 100 aa discarded using CD-HIT (83). Sequences from the reference data sets (UniProt and 1° cohort genomes) were then combined and aligned using MAFFT (82), and discordant sequences were removed using MaxAlign and then the Lake Erken metagenome sequences were added and realigned to construct phylogenetic trees using the neighbor-joining (NJ) method with WAG substitution model and estimated alpha. Trees were visualized in Dendroscope (88). Putative phylogeny of metagenomic sequences was inferred by branch placement directly adjacent to reference sequence clusters.

**Data availability.** Time series MiSeq sequences were deposited in ENA (PRJEB20109) under accession numbers ERS1884053 to ERS1884254. Mixed-culture sequences are available from ENA PRJEB35605. 454 data used in the time series are available at SRA under accession number SRR097418 (https://trace.ncbi.nlm.nih.gov/Traces/sra/?run=SRR097418) (11). HiSeq metagenome sequences are available at ENA project PRJEB37497 under accession numbers ERS4415328 to ERS4415335, and the assembled metagenomes are available at IMG (https://img.jgi.doe.gov/cgi-bin/m/main.cgi) with Taxon Object IDs 3300020141, 3300020151, 3300020159, 3300020160, 3300020161, 3300020162, 3300020172, and 3300020205. Rscripts used for correlation and network analyses are available from https://github.com/rmondav/publications.

## SUPPLEMENTAL MATERIAL

Supplemental material is available online only.
**FIG S1**, EPS file, 0.9 MB.
**FIG S2**, EPS file, 0.4 MB.
**TABLE S1**, PDF file, 0.03 MB.
**TABLE S2**, PDF file, 0.03 MB.

## ACKNOWLEDGMENTS

R.M., sequencing of the metagenomes, and publication were supported by grants in 2014 and 2015 from the Malméns Stiftelsen. S.L.G., sequencing of the mixed cultures were supported by a SciLifeLab Fellowship, and Kungl. Vetenskapsakademiens stiftelser grant BS2017-0044, Alexander Eiler's Swedish Research Council (VR) grant 2012-4592, and Lars Tranvik's Knut and Alice Wallenberg Foundation grant KAW 2013.0091. S.B. was supported by grants from the Swedish Research Council (VR) and the Swedish Research Council Formas. The sequencing of the Lake Erken 16S rRNA time series data set and chemical data was supported by the Swedish Infrastructure for Ecosystem Science (SITES). The funders had no role in study design, data collection and interpretation, or the decision to submit the work for publication. We gratefully acknowledge the computing resources provided by SNIC through the Uppsala Multidisciplinary Centre for Advanced Computational Science (UPPMAX) under UPPNEX projects 2015047, 2016272, and 2017147 and sequencing infrastructure support from the SciLifeLab National Genomics Infrastructure.

The sequencing of the Lake Erken 16S rRNA time series data set and chemical data was done with technical assistance from Omneya Ahmed, Pilar López Hernández, Helena Enderskog, Erika Bridell, and Kristiina Mustonen.

S.B., S.L., E.S.L., and R.M. contributed to the genomic sampling design, organization, and selection. S.L.G. and M.B. designed and implemented the mixed-culture experiments. R.M. designed and performed the bioinformatic analyses. R.M. and S.L.G. synthesized the conceptual framework which was drafted by R.M. in consultation with S.L.G., with editorial contributions from all authors.

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
