## [Reviewer comments · mSystems]

Streamlined and abundant bacterioplankton thrive in functional cohorts

Rhiannon Mondav, Stefan Bertilsson, Moritz Buck, silke langenheder, Eva Lindstrom, and Sarahi Garcia

Corresponding Author(s): Rhiannon Mondav, Uppsala University

Review Timeline:

Submission Date:	April 14, 2020
Editorial Decision:	May 22, 2020
Revision Received:	August 19, 2020
Accepted:	September 11, 2020

Editor: Rup Lal

Reviewer(s): Disclosure of reviewer identity is with reference to reviewer comments included in decision letter(s). The following individuals involved in review of your submission have agreed to reveal their identity: Sarah Ben Maamar (Reviewer #2)

Transaction Report:

DOI: <https://doi.org/10.1128/mSystems.00316-20>

May 22, 2020

Dr. Rhiannon Mondav
Uppsala University
Uppsala
Sweden

Re: mSystems00316-20 (Streamlined freshwater bacterioplankton Nanopelagicales (acl) and Ca. Fonsibacter (LD12) thrive in functional cohorts)

Dear Dr. Rhiannon Mondav:

The reviewers have now commented on your manuscript and they are advising for modification prior acceptance. The journal staff has reported the presence of >10 supplemental materials. The authors would need to reduce prior resubmission. Please refer to instructions to authors for mSystems.

Supplemental material can be posted by mSystems{trade mark, serif} or, if authors prefer, can be submitted by the authors for posting by a third-party service such as Dryad, figshare, or a similar repository. In the latter case, the assigned accession number(s) must be included in the manuscript submitted for review. Supplemental material will be peer reviewed along with the manuscript and must be uploaded to the eJournalPress (eJP) peer review system at the initial manuscript submission. For initial submission, this material must be uploaded as a single PDF. At the modification stage, however, each item in the supplemental material must be submitted as a separate file; i.e., multiple figures should not be zipped together or combined in a single PDF. ASM will post no more than 10 individual supplemental items. The maximum size permitted for an individual file is 3 MB (20 MB for movie or Excel data set files).

Below you will find the comments of the reviewers.

To submit your modified manuscript, log onto the eJP submission site at <https://msystems.msubmit.net/cgi-bin/main.plex>. If you cannot remember your password, click the "Can't remember your password?" link and follow the instructions on the screen. Go to Author Tasks and click the appropriate manuscript title to begin the resubmission process. The information that you entered when you first submitted the paper will be displayed. Please update the information as necessary. Provide (1) point-by-point responses to the issues raised by the reviewers as file type "Response to Reviewers," not in your cover letter, and (2) a PDF file that indicates the changes from the original submission (by highlighting or underlining the changes) as file type "Marked Up Manuscript - For Review Only."

Due to the SARS-CoV-2 pandemic, our typical 60 day deadline for revisions will not be applied. I hope that you will be able to submit a revised manuscript soon, but want to reassure you that the

journal will be flexible in terms of timing, particularly if experimental revisions are needed. When you are ready to resubmit, please know that our staff and Editors are working remotely and handling submissions without delay. If you do not wish to modify the manuscript and prefer to submit it to another journal, please notify me of your decision immediately so that the manuscript may be formally withdrawn from consideration by mSystems.

To avoid unnecessary delay in publication should your modified manuscript be accepted, it is important that all elements you upload meet the technical requirements for production. I strongly recommend that you check your digital images using the Rapid Inspector tool at <http://rapidinspector.cadmus.com/RapidInspector/zmw/>.

Sincerely,

Rup Lal

Editor, mSystems

Journals Department
Reviewer comments:

Reviewer #1 (Comments for the Author):

In this study, Mondav et al. investigate the ecological functioning of highly abundant candidate genera from Actinobacteria and Proteobacteria found in freshwater communities. They use a combination of culture and 16S/metagenomic sequencing to study the ecological relationships between these groups at both a taxonomic and functional level. The study is interesting and provides useful theoretical insights into the cooperative strategies and ecology of aquatic microbial communities. I find that a particularly strong aspect of the work is the combination of culture-dependent and independent methodologies to obtain complementary evidence for some of their findings.

1) My main suggestion is to improve the readability/presentation of the results. As the manuscript stands, the Results are entirely descriptive and make it hard to understand the rationale behind each approach taken. I liked the Discussion section as it better conveys the meaning behind their findings. Either merging Results and Discussion together or integrating some of the Discussion aspects into the Results and then making a shorter Discussion would improve overall readability.

2) Somewhat related to the above point, it is not entirely clear to me why the catalase katG was the primary focus in assessing why Nanopelagicus species grew better in culture when compared to freshwater environments. Wouldn't it have made more sense to take an agnostic approach and look for differences in gene content/SNV variation/expression between genomes in the two conditions and identify the targets that way? Given the authors have metagenomic data available, this warrants further investigation. The authors also discuss the "low functionality" of this gene in Nanopelagicales. Without gene expression or protein activity data this seems highly speculative and should be toned down or further verified.

3) I also think that the Figures could be substantially improved. For instance, given their complex study design (culturing, sequencing, multiple time points, etc.) it would be beneficial to include a schematic at least as a Supplementary Figure summarizing their study and their methodological approach. This could even be built from the map included in Fig. S7. A simplified version of Figure 1 (with fewer nodes or labels) focusing on the most interesting aspects the authors want to highlight would also help.

Reviewer #2 (Comments for the Author):

The manuscript "Streamlined freshwater bacterioplankton Nanopelagicales (acI) and "Ca. Fonsibacter" (LD12) thrive in functional cohorts" describes and analyses the dynamics over 8 years of mainly three bacterioplanktonic genera Actinobacterial genera "Ca. Planktophila" (acI-A) and "Ca. Nanopelagicus" (acI-B), and the Proteobacterial genus "Ca. Fonsibacter" (LD12) and their primary cohorts, in the dimictic Lake Erken (Sweden).

Through their cultivation and sequencing approaches, combined with the taxa correlation network analysis, the authors found that "Ca. Planktophila" (acI-A) and "Ca. Nanopelagicus" (acI-B) are likely relying on a common good produced by the cohort allowing detoxification (catalase catalase-peroxidase enzyme KatG) while "Ca. Fonsibacter" would be mainly relying on products of the degradation of Stramenopiles cells.

Overall the research is interesting thanks to its original approach on networks and to its good connection to ecological concepts.

The way the authors used some elements of network theory to interpret their correlation network is rather innovative, although I think the authors could have gone further there.

The authors could have indeed included centrality measures of their networks, looked at network motifs or activity motifs, and could have attempted to measure the distance between their networks as well.

But the main remarks I have are regarding the readability of the MS. I found hard to follow the logic in the results section until I came to the Discussion.

The Results section is very long and way too detailed and the subtitles are not informative. One way to improve the readability of the paper is to use subtitles appropriately: make them more explicit about what is/are the main take away of each subsection. The part of the results focusing on katG should also be in its own subsection.

Same comment for the subtitles of the Discussion section, which are basically the same.

Another point is about the network figures, they are way too crowded. Is there any way for the authors to only keep relevant names of microbes, such as only those cited in the MS and actually helpful to understand the main findings?

Also, the supplementary material is very dense and some figures would have deserved to be part of the main paper, as the some of the ones related to katG since this gene is part of the main funding. On a related note, I think the authors should also give more info about why and how they suspected this gene could help them cultivating "Ca. Planktophilia" (acl-A) and "Ca. Nanopelagicus" (acl-B). I assumed this was based on the literature but the reasons for the focus on katG deserves to be better explained in my opinion.

The authors need to introduce the notion of assortativity in a network and explain how this is evaluated for non familiar readers.

I also think the authors are sometimes a too assertive in their interpretation of the results in the Discussion (ex. L284-286; L294-295; L364-366). The authors use a lot the word "support", which differs from "suggest".

L-295-296: "High assortativity in the mixed cultures at all levels except phylum, supports that niche differentiation contributed to co-occurrence in the cultures." High assortativity can be explained by a lot of other phenomena besides niche differentiation (predation, cooperation, etc..) Please be careful in your statement and reformulate.

Some of the hypothesis formulated by the authors could have been tested and would have been of great value for the MS. For example, the authors could have attempted to measure the expression of katG through qPCR and KatG activity with enzymatic assays to confirm its high or low activity. Also, the MS would have really benefited from having some reconstructed genomes from the metagenomes to show the loss of potentially strategic genes.

Finally the MS really deserves a better conclusion with some insights on the implications of this work, its limits AND some perspectives for future work.

Typo mistakes:

L197: replace "genera" by genus

L199: replace "Nanopegicales" by Nanopelagicales

L209-211: font is different from the rest of the MS

L338-344: sentence is too long, divide the sentence in 2

Fig 2 caption: replace "aand" by and

Response to Reviewer comments:

**please note that line numbers refer to the marked-up-manuscript for reviewers and not the clean version.

Reviewer #1

1) *My main suggestion is to improve the readability/presentation of the results. As the manuscript stands, the Results are entirely descriptive and make it hard to understand the rationale behind each approach taken. I liked the Discussion section as it better conveys the meaning behind their findings. Either merging Results and Discussion together or integrating some of the Discussion aspects into the Results and then making a shorter Discussion would improve overall readability.*

-Sections from the methods and discussion have been moved to results and the rationale behind decisions and approaches has been expanded. We believe this has massively improved the readability of results and we thank the reviewer for pointing this problem out to us. Please see L168-169, L175-176, L192-195, L312-319, and L391-399.

2) *Somewhat related to the above point, it is not entirely clear to me why the catalase *katG* was the primary focus in assessing why *Nanopelagicus* species grew better in culture when compared to freshwater environments. Wouldn't it have made more sense to take an agnostic approach and look for differences in gene content/SNV variation/expression between genomes in the two conditions and identify the targets that way? Given the authors have metagenomic data available, this warrants further investigation. The authors also discuss the "low functionality" of this gene in *Nanopelagicales*. Without gene expression or protein activity data this seems highly speculative and should be toned down or further verified.*

-We have now separated out the section on KatG (L310-388) to recognise the importance of this enzyme in culturability of *Nanopelagicales*. We did look at differences in gene detection in the lake metagenomes and published genomes and we hope our increased descriptions show this more clearly (L349-351 and L364-371). However it was 1) never our intention to do transcriptome or proteomic work in the (granted) project design, and 2) the information about KatG became available three years after the data for this project was gathered while both the first and senior authors were on parental leave. We decided it was better to enfold this important finding into our work as best we could given the retroactive nature of the analyses than to pretend that the solution to axenic culture of many *Nanopelagicales* had not been found. We thank the reviewer for pointing out that the rationale behind our intense scrutiny of *katG* and KatG is unclear so we have increased the description of decisions and background information (L312-319). We believe this has resulted in clearer demonstration of the connection between known crystallography and known activity assay data (ie evidence of high versus low functionality) and our sequences. We would be interested in gathering more activity or crystallography data on this enzyme but that would be a different project for the future.

-We have also now included a more holistic approach to the network and growth dependencies rather than focus solely on KatG, please see the sub-section "Metabolic dependencies clarified inter-specific relationships" (L390-422).

3) *I also think that the Figures could be substantially improved. For instance, given their complex study design (culturing, sequencing, multiple time points, etc.) it would be beneficial to include a schematic at least as a Supplementary Figure summarizing their study and their methodological approach. This could even be built from the map included in Fig. S7.*

A simplified version of Figure 1 (with fewer nodes or labels) focusing on the most interesting aspects the authors want to highlight would also help.

-Thank you for the guidance, we direct your attention to the new Fig. 2. It is a composite of several original figures into an overall schematic of this project. We are grateful for this nudge and we believe this new figure greatly improves the readability of the entire manuscript. Cheers!

-We have altered the original Figure 1 (now Fig. 4) by separating out the four 1° cohorts which improves clarity, and also added Venn diagram (Fig. 7) which we believe more clearly shows the information we were trying to convey. We have retained the complete network figures in the supplementary information.

Reviewer #2

Overall the research is interesting thanks to its original approach on networks and to its good connection to ecological concepts. The way the authors used some elements of network theory to interpret their correlation network is rather innovative, although I think the authors could have gone further there. The authors could have indeed included centrality measures of their networks, looked at network motifs or activity motifs, and could have attempted to measure the distance between their networks as well.

-We thank you for your time and thought on reading our manuscript and the general suggestions you have made. We have now included the details of other network analyses we did that were 'null results' to give a more holistic picture of our work and findings they are uploaded as 'null results for reviewers tables 1 and 2'. We have also added a new figure (Fig. 7) which is a Venn diagram of the metabolic complementarity of the 1° cohorts to better convey findings in line with your suggestion. We did however decide not to measure distance between the 1° cohort networks as 1) the other stats suggested it would also be a null result and 2) a somewhat meaningless analysis given the degree of data manipulation involved in separating out 1° cohort networks. I will however keep distance measurements in mind for future work. Thank you for your suggestions they inspired us to improve the manuscript.

But the main remarks I have are regarding the readability of the MS. I found hard to follow the logic in the results section until I came to the Discussion. The Results section is very long and way too detailed and the subtitles are not informative. One way to improve the readability of the paper is to use subtitles appropriately: make them more explicit about what is/are the main take away of each subsection.

-Unfortunately, in order to accommodate reviewer and co-author suggestions, we have increased the length of the results. However, we hope that the improved readability compensates for this. We have replaced the section subheadings with informative subtitles as per your suggestion and believe it has improved the readability. Thank you. Please see L166, 187, 231, 239, 249, 270, 288, 310, and 390.

-We also fixed the sub-heading problem in the discussion section, thank you! L426, 445, 470, and 530.

-We have also greatly increased the described rationale behind approaches. Please see L168-169, L175-176, L192-195, L312-319, and L391-399. We are grateful for your suggestions.

The part of the results focusing on katG should also be in its own subsection.

-Done! We have now separated out the section on KatG (L310-388) to recognise the importance of this enzyme in culturability of Nanopelagicales.

Another point is about the network figures, they are way too crowded. Is there any way for the authors to only keep relevant names of microbes, such as only those cited in the MS and actually helpful to understand the main findings?

-The 1^o cohort network figure was already significantly reduced from the whole network, but in response we have separated each of the cohort networks for the time-series (see Fig. 4) but kept original in the supplementary section (Fig. S1) for those who want to look at it. We have also removed the names of the target genera from Fig. 4 as they are already identified by colour coding and so the text was superfluous.

Also, the supplementary material is very dense and some figures would have deserved to be part of the main paper, as the some of the ones related to katG since this gene is part of the main funding. On a related note, I think the authors should also give more info about why and how they suspected this gene could help them cultivating "Ca. Planktophila" (acI-A) and "Ca. Nanopelagicus" (acI-B). I assumed this was based on the literature but the reasons for the focus on katG deserves to be better explained in my opinion.

--Most of the original supplementary section is either in the main text (now Fig 2, 3, & 6) or deleted (Supplementary Tables).

-We have now separated out the section on KatG to recognise the importance of this enzyme in culturability of Nanopelagicales. We thank the reviewer for pointing out that the rationale behind our intense scrutiny of *katG* and KatG is unclear so we have increased the description of decisions and background information (L312-319). We believe this has resulted in clearer demonstration of the connection between known crystallography and known activity assay data (ie evidence of low functionality) and our sequences. We would be interested in gathering more activity or crystallography data on this enzyme but that would be a different project for the future.

The authors need to introduce the notion of assortativity in a network and explain how this is evaluated for non familiar readers.

-done, please see L194-195

I also think the authors are sometimes a too assertive in their interpretation of the results in the Discussion (ex. L284-286;

-we have added "directly controlled" to tone down the statement. L441-443.

L294-295;

-we believe the statement (L452-453) is acceptable and draw your attention to the words "main" and "success", we do not claim filtering is not a driver of their presence, but that it is not the driver of their success. We have changed "support" to "suggest" (L453-455).

L364-366).

-we have added the word "most" to tone down the statement about dependencies in-line with findings on vitamin-B12 in this study (L547-551). There are publications that outline the defense differentiation (references 3, 32, 49) so we feel confident in the boldness of that component of the statement.

The authors use a lot the word "support", which differs from "suggest".

-we have changed "support" for "how it may affect" L333, "confirms" L451, and "suggests" L517.

L-295-296: "High assortativity in the mixed cultures at all levels except phylum, supports that niche differentiation contributed to co-occurrence in the cultures." High assortativity can be explained by a lot of other phenomena besides niche differentiation (predation, cooperation, etc...) Please be careful in your statement and reformulate.

-we have revised this sentence and appreciate the reviewer bringing this to our attention. (L454-457)

Some of the hypothesis formulated by the authors could have been tested and would have been of great value for the MS. For example, the authors could have attempted to measure the expression of katG through qPCR and KatG activity with enzymatic assays to confirm its high or low activity.

-It was 1) never our intention to do transcriptome or proteomic work in the (granted) project design, and 2) the information about KatG became available three years after the data for this project was gathered while both the first and senior authors were on parental leave. Samples etc were no longer available to retroactively do such work. We would be interested in gathering more activity or crystallography data on this enzyme but that would be a different project with different funding for the future. If the reviewer is interested in collaborating on this project we would be interested in hearing from them.

Also, the MS would have really benefited from having some reconstructed genomes from the metagenomes to show the loss of potentially strategic genes.

-we actually did recover many MAGs and analysed them after annotation, however we did not include this information in the original manuscript for reasons described in the revised version. Please see L322-327.

Finally the MS really deserves a better conclusion with some insights on the implications of this work, its limits AND some perspectives for future work.

-We have worked on the discussion section to increase its breadth, connection to ecological theory, and also added several suggestions for future work. Thank you for kindly pointing out the discussion was letting the rest of the manuscript down. Please see L502-511, L526-531, L554-557.

Typo mistakes:

L197: replace "genera" by genus

L199: replace "Nanopegicales" by Nanopelagicales

L209-211: font is different from the rest of the MS

L338-344: sentence is too long, divide the sentence in 2

Fig 2 caption: replace "aand" by and

-thank you !

September 11, 2020

Dr. Rhiannon Mondav
Uppsala University
Uppsala
Sweden

Re: mSystems00316-20R1 (Streamlined and abundant bacterioplankton thrive in functional cohorts)

Dear Dr. Rhiannon Mondav:

Your manuscript has been accepted, and I am forwarding it to the ASM Journals Department for publication. For your reference, ASM Journals' address is given below. Before it can be scheduled for publication, your manuscript will be checked by the mSystems senior production editor, Ellie Ghatineh, to make sure that all elements meet the technical requirements for publication. She will contact you if anything needs to be revised before copyediting and production can begin. Otherwise, you will be notified when your proofs are ready to be viewed.

Sincerely,

Rup Lal
Editor, mSystems

Journals Department
Fig. S2: Accept
Table S1: Accept
Table S2: Accept
Fig. S1: Accept